# Stein Diffusion Guidance: Training-Free Posterior Correction for Sampling Beyond High-Density Regions

**Van Khoa Nguyen** [1 2]  **Lionel Blondé** [1]  **Alexandros Kalousis** [1]

## Abstract

Training-free diffusion guidance offers a flexible framework for leveraging off-the-shelf classifiers without additional training. Yet, current approaches hinge on posterior approximations via Tweedie's formula, which often yield unreliable guidance, particularly in low-density regions. Stochastic optimal control (SOC), in contrast, enables principled posterior sampling but remains computationally prohibitive for efficient inference. In this work, we reconcile the strengths of these paradigms by introducing Stein Diffusion Guidance (SDG), a novel training-free framework grounded in a surrogate SOC objective. We establish a new theoretical bound on the SOC value function, revealing the necessity of correcting approximate posteriors to reflect true diffusion dynamics. Building on Stein variational inference, SDG computes the steepest descent direction that minimizes the Kullback-Leibler divergence between approximate and true posteriors. By integrating a principled Stein correction mechanism along with a novel running cost functional, SDG enables effective guidance in low-density regions. Our experiments on diverse image-guidance tasks and on challenging small-ligand sampling for protein docking suggest that SDG consistently outperforms standard training-free guidance methods and highlights its potential for broader posterior sampling problems beyond high-density regimes.

## 1. Introduction

In many scientific domains, key discoveries often depend on identifying rare samples buried within large data distributions. For instance, while billions of molecules exist in

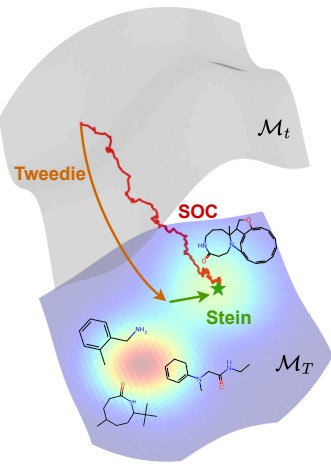

*Figure 1.* SDG provides a computationally efficient, training-free alternative framework to SOC-based diffusion posterior sampling for molecule guidance tasks in low-density regions. We hypothesize that sampling from low-density regions facilitates the discovery of more nontrivial, complex molecules that satisfy desired docking constraints across different experimental target proteins.

chemistry (Polishchuk et al., 2013), only a minute fraction possesses properties relevant to drug discovery. We posit that such high-value samples often reside in low-density regions, making their identification both challenging and prone to error. This challenge has fueled growing interest in methods that accelerate the search for rare, property-rich samples. Generative methods, particularly diffusion models (Ho et al., 2020; Song et al., 2021), have demonstrated strong performance in modeling complex, high-dimensional distributions. However, when trained on unlabeled data, diffusion models predominantly sample from high-density data regions, thereby overlooking the low-density areas where high-value samples are likely to exist. This limitation hinders their effectiveness in tasks that require discovery beyond high-density regions. Numerous studies have been proposed to address this challenge. A particular class of methods leverages an auxiliary classifier (Dhariwal & Nichol, 2021; Sehwag et al., 2022; Lee et al., 2023) to guide pre-trained diffusion models toward regions of interest. However, classifier-based diffusion guidance introduces additional complexity by requiring classifiers trained on multiple noise levels. Moreover, recent studies suggest that noisy classifier gradients can misguide samples, causing

[1]HES-SO Geneva [2]Department of Computer Science, University of Geneva, 1205 Geneva, Switzerland. Correspondence to: Van Khoa Nguyen <van-khoa.nguyen@etu.unige.ch>.

*Proceedings of the 43rd International Conference on Machine Learning*, Seoul, South Korea. PMLR 306, 2026. Copyright 2026 by the author(s).

them to fall off generative manifolds (Chung et al., 2023; Guo et al., 2024). This issue can be even more severe in low-data regimes, where diffusion models are already less accurate, and gradients tend to be less reliable (Sehwag et al., 2022).

Stochastic optimal control (SOC) (Nüsken & Richter, 2021; Domingo-Enrich et al., 2024) has recently been explored to fine-tune diffusion models for a variety of downstream tasks (Uehara et al., 2024; Wang et al., 2025; Domingo-Enrich et al., 2025). These approaches steer the diffusion process towards desired targets by incorporating an auxiliary controller into the stochastic differential equation (SDE) that governs the generative reverse diffusion dynamics. Uehara et al. (2024) further relates SOC to classifier-based guidance, where the reward functions are classifiers trained on clean data, which are readily available across many domains. This enables SOC-based diffusion guidance to leverage off-the-shelf classifiers directly. However, computing the optimal control value requires backpropagating reward signals through entire neural-SDE sampling trajectories (Tzen & Raginsky, 2019; Uehara et al., 2024), which presents a significant drawback restricting the scalability and practicality of SOC-based sampling methods. To circumvent this problem, recent works have proposed approximating the diffusion posterior through Tweedie's formula (Robbins, 1992). This avenue has been primarily explored in the contexts of general inverse problems (Chung et al., 2023; MOUFAD et al., 2025), and image diffusion guidance applications (Yu et al., 2023; Ma et al., 2024; Rout et al., 2025; Janati et al., 2025; Dinh et al., 2025). These methods are often referred to as *training-free diffusion guidance*, as they leverage off-the-shelf classifiers without requiring additional training across noise levels. However, the posterior approximation via Tweedie's formula is biased and inherently suboptimal, which frequently leads to unreliable guidance, particularly in low-density data regions.

Here, we summarize our contributions: (i) we propose a low-density diffusion guidance framework formulated under stochastic optimal control, which introduces a novel cost-to-go function; (ii) we theoretically prove that approximating the diffusion posterior via Tweedie's formula is inferior and requires further correction steps; (iii) we introduce a Stein correction mechanism for SOC-based diffusion posterior sampling, which leverages Stein variational inference tools (Liu & Wang, 2016; Liu, 2017) to iteratively minimize the Kullback-Leibler (KL) divergence between approximate and true posteriors. Our experimental results on diverse image guidance tasks and complex ligand-protein docking problems suggest that the proposed Stein correction is critical for efficiently enhancing diffusion posterior sampling, particularly in low-density regions, leading to effective training-free diffusion guidance and the discovery of more high-affinity ligands across different target proteins.

## 2. Preliminaries

### 2.1. Diffusion Models

Song et al. (2021) introduce a continuous-time, continuous-state diffusion framework, $\forall t \in [0, T]$, $\mathbf{x} \in \mathbb{R}^d$, utilizing a pair of forward and backward SDEs. The forward process diffuses data samples, $\mathbf{x}_T \sim p_T$, towards an easy-to-sample prior, $\mathbf{x}_0 \sim p_0$. Its dynamic is the solution to an Itô SDE: $d\mathbf{x}_t = \mathbf{b}(\mathbf{x}_t, t)dt + \sigma(t)d\mathbf{w}$, where $dt < 0$ is an infinitesimal timestep, $\mathbf{w}$ is the Weiner process, and $\mathbf{b}(\mathbf{x}_t, t), \sigma(t)$ denote the drift and diffusion coefficient, respectively. A backward process gradually denoises samples from the prior $p_0$ and back to the data distribution $p_T$; the reverse dynamic corresponds to another Itô SDE of the form:

$$\mathbb{P}: d\mathbf{x}_t = \left(-\mathbf{b}(\mathbf{x}_t, t) + \sigma(t)^2 \nabla_{\mathbf{x}_t} \log p_t(\mathbf{x}_t)\right)dt + \sigma(t)d\mathbf{w} \tag{1}$$

where the marginal data score, $\nabla_{\mathbf{x}_t} \log p_t(\mathbf{x}_t)$, can be estimated by the score-matching technique (Hyvärinen & Dayan, 2005) via a time-dependent score-based network, $\mathbf{s}_\theta(\mathbf{x}_t) \approx \nabla_{\mathbf{x}_t} \log p_t(\mathbf{x}_t)$; for simplicity, we omit the explicit $t$-dependence in the score model notation. Unlike Song et al. (2021), we employ a positive infinitesimal reverse timestep to facilitate our theoretical development of novel diffusion guidance in subsequent sections. In sampling, diffusion models can incorporate a classifier $r(\cdot)$ to guide samples toward regions with desired properties, a technique known as *classifier-based diffusion guidance* (Dhariwal & Nichol, 2021). The conditional score can be factorized into $\nabla_{\mathbf{x}_t} \log p_t(\mathbf{x}_t|y) \propto \nabla_{\mathbf{x}_t} \log p_t(\mathbf{x}_t) + \nabla_{\mathbf{x}_t} r(y|\mathbf{x}_t)$, which requires training the classifier $r(\cdot)$ on noisy data of multiple noise levels, $\mathbf{x}_t \sim p_t(\mathbf{x}_t|\mathbf{x}_T), \forall t \in [0, T]$.

### 2.2. Stein Variational Gradient Descent

Variational inference (VI) approximates a complex target distribution $p$ using a simpler distribution $q$ from a tractable family $\mathcal{Q}$ of distributions by minimizing the KL divergence, $\arg\min_q D_{KL}(q\|p)$. Stein variational gradient descent (SVGD) (Liu & Wang, 2016) provides a nonparametric approach, representing $q$ as a set of interacting particles that are deterministically evolved along a direction $\phi$ to most efficiently decrease the KL divergence. The following lemma summarizes the core results:

**Lemma 2.1.** *Let $T_\varepsilon(x) = x + \varepsilon\phi(x)$ be a smooth invertible map for small $\varepsilon$ and denote $q_\varepsilon = (T_\varepsilon)_\# q$. Among all vector fields $\phi$ in the unit ball of a vector-valued RKHS $\mathcal{H}^d$, i.e. $\|\phi\|_{\mathcal{H}^d} \leq 1$, the direction of steepest descent of the Kullback–Leibler divergence $D_{\mathrm{KL}}(q_\varepsilon\|p)|_{\varepsilon=0}$ is given by*

$$\phi^* = \arg\max_{\phi \in \mathcal{H}^d, \|\phi\|_{\mathcal{H}^d} \leq 1} \mathbb{E}_{x \sim q}\left[\mathrm{tr}\left(\mathcal{A}_p\phi(x)\right)\right]$$

*with the Stein operator $\mathcal{A}_p$ is defined as*

$$\mathcal{A}_p\phi(x) = \nabla_x \log p(x)\,\phi(x)^\top + \nabla_x\phi(x)$$

where $\phi$ is the Stein class of the target density $p$. Liu et al. (2016) solve the problem 2.1 as kernelized Stein discrepancy (KSD) that yields $\phi^*(\mathbf{x}^i) = \mathbb{E}_{\mathbf{x}^j \sim q(\mathbf{x})}\left[\nabla_{\mathbf{x}^j} \log p(\mathbf{x}^j)k(\mathbf{x}^i, \mathbf{x}^j) + \nabla_{\mathbf{x}^j} k(\mathbf{x}^i, \mathbf{x}^j)\right]$, where $k(\mathbf{x}^i, \mathbf{x}^j)$ is Radial Basic Function (RBF) kernel $k(\mathbf{x}^i, \mathbf{x}^j) = \exp\left(-\frac{1}{m}\|\mathbf{x}^i - \mathbf{x}^j\|_2^2\right)$ with a bandwidth $m$ typically set using the heuristic $m = med(\|\mathbf{x}^i - \mathbf{x}^j\|_2^2)/\log N$, based on the squared pairwise distance median (med) among $N$ particles. SVGD can approximate the intractable target density $p$ by gradually transporting a set of $N$ initial particles $\{\mathbf{x}_0^i\}_{i=0}^N \sim q_0$ along the direction $\phi^*$, which relies on the computable score $\nabla_{\mathbf{x}} \log p(\mathbf{x})$. The first term of $\phi^*$ represents a kernel-weighted gradient ascent direction that pushes the particles toward the high-density regions. The second term denotes the repulsive force that prevents the particles from collapsing into the local modes of $p(\mathbf{x})$. In practice, the set of initial particles can be set arbitrarily, and the theoretical convergence of particles to their actual density can be guaranteed $\sum_{i=1}^N h(\mathbf{x}^i)/N - \mathbb{E}_{q(\mathbf{x})}[h(\mathbf{x})] = \mathcal{O}(1/\sqrt{N})$ for bounded testing functions $h$ (Del Moral, 2013).

### 2.3. Stochastic Optimal Control

Stochastic optimal control (SOC) (Nüsken & Richter, 2021) seeks an optimal controller that steers the behavior of a given stochastic system to minimize a pre-specified cost function. For the stochastic dynamical diffusion system in Equation 1, we formalize an affine-control problem as follows:

$$\inf_{\mathbf{u} \in \mathcal{U}} \mathbb{E}\left[\int_t^T \left(\frac{1}{2}\|\mathbf{u}(\mathbf{x}_s^{\mathbf{u}}, s)\|^2 + f(\mathbf{x}_s^{\mathbf{u}}, s)\right)ds + g(\mathbf{x}_T^{\mathbf{u}})\right]$$

$$\text{s.t.} \quad \mathbb{P}^{\mathbf{u}}: \quad d\mathbf{x}_t^{\mathbf{u}} = \left(-\mathbf{b}(\mathbf{x}_t^{\mathbf{u}}, t) + \sigma(t)^2 \nabla_{\mathbf{x}_t^{\mathbf{u}}} \log p_t(\mathbf{x}_t^{\mathbf{u}})\right.$$
$$\left. + \sigma(t)\mathbf{u}(\mathbf{x}_t^{\mathbf{u}}, t)\right)dt + \sigma(t)d\mathbf{w}$$
$$(2)$$

where the feedback control $\mathbf{u} : \mathbb{R}^d \times [0, T] \mapsto \mathbb{R}^d$ drives the system dynamics, $f : \mathbb{R}^d \times [t, T] \mapsto \mathbb{R}$ specifies the state cost, $g : \mathbb{R}^d \mapsto \mathbb{R}$ is the terminal cost, and $\mathbb{P}^u$ denotes the controlled probability path measure induced from $\mathbb{P}$. The control objective minimizes the cost functional $J(\mathbf{u}, \mathbf{x}, t) = \mathbb{E}_{\mathbb{P}^{\mathbf{u}}}\left[\int_t^T \left(\frac{1}{2}\|\mathbf{u}(\mathbf{x}_s^{\mathbf{u}}, s)\|^2 + f(\mathbf{x}_s^{\mathbf{u}}, s)\right)ds + g(\mathbf{x}_T^{\mathbf{u}})|\mathbf{x}_t = \mathbf{x}\right]$, whose minimum defines the value function or optimal cost-to-go (Fleming & Soner, 2006), $V(\mathbf{x}, t) = \inf_{\mathbf{u} \in \mathcal{U}} J(\mathbf{u}, \mathbf{x}, t)$. Moreover, verification theorem (Fleming & Soner, 2006; Pham, 2009) relates the optimal control and value function via $\mathbf{u}^* = -\sigma\nabla_{\mathbf{x}}V$. The affine stochastic control problem with a quadratic cost is closely connected to an iterative diffusion optimization using a relative entropy loss (Powell, 2021; Kappen et al.,

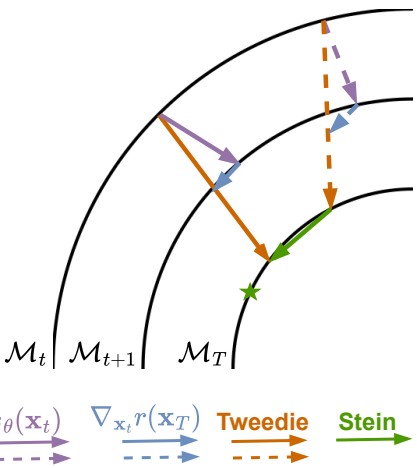

$\mathcal{M}_t$ | $\mathcal{M}_{t+1}$ | $\mathcal{M}_T$

$s_\theta(\mathbf{x}_t)$    $\nabla_{\mathbf{x}_t} r(\mathbf{x}_T)$    **Tweedie**    **Stein**

*Figure 2.* Back-and-forth Stein correction: Noisy particles are first mapped backward to the data manifold, $\mathcal{M}_T$, to obtain approximate posterior samples, which are corrected via the Stein correction and then mapped forward to the current diffusion manifold, $\mathcal{M}_t$, for reward-based low-density diffusion sampling. Dashed arrows indicate standard training-free diffusion guidance methods, while solid arrows represent SDG.

2012; Hartmann & Schütte, 2012), which involves simulating multiple controlled trajectories, computing their cumulative costs, and backpropagating through the trajectories to update a parameterized controller; a more extensive treatment of SOC problem can be found in (Nüsken & Richter, 2021; Domingo-Enrich et al., 2024). Here, we summarize SOC's fundamental theoretical results as follows:

**Lemma 2.2.** *Assuming the state cost $f(\mathbf{x}, s)$ and the terminal cost $g(\mathbf{x})$ have continuous first-order derivatives in space, i.e. $f \in C^1(\mathbb{R}^d \times [t, T]; \mathbb{R})$ and $g \in C^1(\mathbb{R}^d; \mathbb{R})$, the optimal controller $\mathbf{u}^*(\mathbf{x}, t)$ and the value function $V(\mathbf{x}, t)$ for the stochastic optimal control problem (2) are given by:*

$$\mathbf{u}^*(\mathbf{x}, t) = -\sigma(t)\nabla_{\mathbf{x}}V(\mathbf{x}, t)$$
$$V(\mathbf{x}, t) = -\log \mathbb{E}_{\mathbb{P}}\left[\exp\left(-\int_t^T f(\mathbf{x}_s, s)ds - g(\mathbf{x}_T)\right)|\mathbf{x}_t = \mathbf{x}\right]$$

As observed, obtaining the value function requires integrating diffusion trajectories from $\mathbf{x}_t = \mathbf{x}$ under $\mathbb{P}$, and computing the gradients $\nabla_{\mathbf{x}}V(\mathbf{x}, t)$ within these simulation trajectories to recover the optimal controller. Both operations are computationally expensive and substantially slow for practical applications. Moreover, the state cost is often omitted, i.e, $f(\mathbf{x}_s, s) = 0 \,\forall s$. Under this setting, Uehara et al. (2024) establish a connection with classifier-based diffusion guidance, $\mathbf{u}^* = \sigma(t)\nabla_{\mathbf{x}} \log \mathbb{E}_{\mathbb{P}}\left[\exp\left(-g(\mathbf{x}_T)\right)|\mathbf{x}_t = \mathbf{x}\right] \propto \nabla_{\mathbf{x}}\mathbb{E}_{\mathbb{P}}\left[r(\mathbf{x}_T)|\mathbf{x}_t = \mathbf{x}\right]$, wherein $r = -g$, due to the minimization problem, corresponds to an off-the-shelf classifier or a differential reward model.

# 3. Stein Diffusion Guidance

We introduce a novel training-free diffusion guidance framework derived from a surrogate stochastic optimal control formulation. This section is organized as follows. Section 3.1 presents a new SOC cost functional that enables diffusion models to explore in low-density regions. Section 3.2 establishes a variational bound on the SOC value function, showing that existing training-free guidance methods require posterior correction. Section 3.3 proposes a back-and-forth Stein correction, a low-cost alternative to SOC that regularizes posterior samples for effective low-density exploration. Detailed proofs are deferred to Appendix B.

## 3.1. Low Density Diffusion Sampling as Stochastic Optimal Control

We consider the controlled reverse SDE (Equation 2) and its associated probability path measure $\mathbb{P}^{\mathbf{u}}$. We introduce a novel cost functional $\widetilde{J}(\mathbf{u}, \mathbf{x}, t)$ that progressively anneals the marginal density $p_t(\mathbf{x}_t)$ of the uncontrolled SDE (Equation 1) under $\mathbb{P}$ to low-density regions:

$$\widetilde{J}(\mathbf{u}, \mathbf{x}, t) = \mathbb{E}_{\mathbb{P}^{\mathbf{u}}}\Big[\int_t^T \Big(\frac{1}{2}\|\mathbf{u}(\mathbf{x}_s^{\mathbf{u}}, s)\|^2$$
$$+ \alpha(s)\log p_s(\mathbf{x}_s^{\mathbf{u}})\delta(s - t)\Big)ds - \beta(t)r(\mathbf{x}_T^{\mathbf{u}})|\mathbf{x}_t = \mathbf{x}\Big]$$
$$(3)$$

where $\alpha(t)$ and $\beta(t)$ denote the schedules controlling low-density annealing and guidance strength, respectively. The term $\delta(s - t)$ is a Dirac delta function of time, which satisfies $\delta(s - t) = 0$ if $s \neq t$, and $\int \delta(s - t)ds = 1$. We formulate low-density diffusion guidance as a stochastic optimal control problem in the following proposition.

**Proposition 3.1.** *Consider the stochastic optimal control problem with the novel functional cost $\widetilde{J}(\mathbf{u}, \mathbf{x}, t)$ defined in Equation 3. By Lemma 2.2, the marginal density $p_t^{\mathbf{u}}(\mathbf{x}_t)$, the value function $V(\mathbf{x}, t)$, and the optimal control $\mathbf{u}^*(\mathbf{x}, t)$ of the controlled-reverse SDE under $\mathbb{P}^{\mathbf{u}}$ are computed as:*

$$p_t^{\mathbf{u}}(\mathbf{x}_t) = \mathbb{E}_{\mathbb{P}}\Big[p_t^{1-\alpha(t)}(\mathbf{x}_t)\exp(\beta(t)r(\mathbf{x}_T))|\mathbf{x}_t\Big]$$
$$V(\mathbf{x}, t) = -\log\frac{p_t^{\mathbf{u}}(\mathbf{x})}{p_t(\mathbf{x})}$$
$$\mathbf{u}^*(\mathbf{x}, t) = \sigma(t)\nabla_{\mathbf{x}}\log\frac{p_t^{\mathbf{u}}(\mathbf{x})}{p_t(\mathbf{x})}$$

The induced marginal density is the product of the annealed density term $p_t^{1-\alpha(t)}$ and the guidance term $\exp(\beta(t)r(\mathbf{x}_T))$. The latter is an un-normalized energy density with the energy function $-\beta(t)r(\mathbf{x}_T)$. In Section 2.3, SOC constitutes a computationally expensive posterior simulation problem to obtain $\mathbf{x}_T$. In particular, deriving the optimal control $\mathbf{u}^*(\mathbf{x}, t)$ requires backpropagating through sampling trajectories (Uehara et al., 2024; Wang et al., 2025).

To alleviate the memory burden, one can leverage Tweedie's formula to compute the approximate posterior mean of $\mathbf{x_T}$ given $\mathbf{x_t}$, $\mathbb{E}[\mathbf{x}_T|\mathbf{x}_t] = ((\mathbf{x}_t + \gamma^2(t)\mathbf{s}_\theta(\mathbf{x}_t, t))/\eta(t)$, assuming a forward kernel $p_{t|T}(\mathbf{x}_t|\mathbf{x}_T) = \mathcal{N}(\eta(t)\mathbf{x}_T, \gamma^2(t)I)$. This one-step posterior estimation has been explored in many prior works. However, using Tweedie's posterior approximation is inherently suboptimal. As illustrated in Figure 1, this approximation significantly deviates from the endpoint sample given by the true SOC simulation. Below, we analyze the sub-optimality of Tweedie-based posterior samples and propose a novel Stein correction mechanism.

## 3.2. Value Function Variational Bound

We consider the target conditional posterior $p_{T|t}(\mathbf{x}_T \mid \mathbf{x}_t)$, defined as the terminal distribution of $\mathbf{x}_t$ under the uncontrolled reverse process $\mathbb{P}$ from Equation 1. By Proposition 3.1, the low-density reward-guided value function can be written as $V(\mathbf{x}, t) = -\log\frac{p_t^{\mathbf{u}}(\mathbf{x})}{p_t(\mathbf{x})}$, where $p_t^{\mathbf{u}}(\mathbf{x}) = \mathbb{E}_{\mathbf{x}_T \sim p_{T|t}(\mathbf{x}_T|\mathbf{x})}\Big[p_t^{1-\alpha(t)}(\mathbf{x})\exp(\beta(t)r(\mathbf{x}_T))|\mathbf{x}_t = \mathbf{x}\Big]$. To ensure computational tractability, we introduce a surrogate objective $\bar{V}(\mathbf{x}, t, q)$ with $q \in \mathcal{Q}$, which serves as an upper bound on the value function $V(\mathbf{x}, t)$.

**Proposition 3.2.** *Consider a proposal distribution $q$ from a traceable family of distributions $\mathcal{Q}$. Then, the value function in Proposition 3.1 admits the following variational bound:*

$$V(\mathbf{x}, t) \leq \bar{V}(\mathbf{x}, t, q)$$
$$= -\mathbb{E}_{\mathbf{x}_T \sim q_{T|t}(\mathbf{x}_T|\mathbf{x})}\Big[\log\Big(p_t^{-\alpha(t)}(\mathbf{x})\exp\left(\beta(t)r(\mathbf{x}_T)\right)\Big)|\mathbf{x}_t = \mathbf{x}\Big]$$
$$+ D_{KL}\Big(q(\mathbf{x}_T|\mathbf{x}_t)\|p(\mathbf{x}_T|\mathbf{x}_t)\Big)\Big|_{\mathbf{x}_t=\mathbf{x}}$$
$$= \alpha(t)\log p_t(\mathbf{x}) - \beta(t)\mathbb{E}_{\mathbf{x}_T \sim q_{T|t}(\mathbf{x}_T|\mathbf{x})}\Big[r(\mathbf{x}_T)|\mathbf{x}_t = \mathbf{x}\Big]$$
$$+ D_{KL}\Big(q(\mathbf{x}_T|\mathbf{x}_t)\|p(\mathbf{x}_T|\mathbf{x}_t)\Big)\Big|_{\mathbf{x}_t=\mathbf{x}}$$

The first term on the RHS drives samples toward low-density regions, while the second term guides them toward regions with desired properties. When posterior samples $\mathbf{x}_T$ are estimated via Tweedie's formula, these two terms together reproduce the training-free diffusion guidance explored in prior works (Chung et al., 2023; MOUFAD et al., 2025; Yu et al., 2023; Ma et al., 2024; Shen et al., 2024; Rout et al., 2025; Janati et al., 2025; Dinh et al., 2025). The last term serves as a KL regularization that minimizes the divergence between the proposal posterior $q(\mathbf{x}_T|\mathbf{x}_t)$ and the true posterior $p(\mathbf{x}_T|\mathbf{x}_t)$. Thus, optimizing only the first two terms of $\bar{V}(\mathbf{x}, t, q)$ under the Tweedie-based posterior approximation is suboptimal, since the resulting controller neglects the KL term and thus fails to remain close to the true posterior. To overcome this inherent limitation, we introduce a Stein correction mechanism that refines the approximate posterior samples before leveraging them for guidance.

### 3.3. Stein Meets Tweedie for Surrogate Stochastic Optimal Control

Given the upper bound of the value function, we derive the optimal controller that minimizes this bound, which we term the *surrogate* stochastic optimal control problem. By Lemma 2.2, the optimal control $\bar{\mathbf{u}}^*(\mathbf{x}_t, t)$ corresponding to the surrogate value function $\bar{V}(\mathbf{x}, t, q)$ is computed as:

$$
\frac{\bar{\mathbf{u}}^*(\mathbf{x}_t, t)}{\sigma(t)} = -\nabla_{\mathbf{x}_t} \bar{V}(\mathbf{x}_t, t, q)
$$

$$
= \underbrace{-\alpha(t)\mathbf{s}_\theta(\mathbf{x}_t) + \beta(t)\nabla_{\mathbf{x}_t}\mathbb{E}_{\mathbf{x}_T \sim q_{T|t}(\mathbf{x}_T|\mathbf{x}_t)}\Big[r(\mathbf{x}_T)\Big]}_{\mathbf{I}}
$$

$$
+ \underbrace{-\nabla_{\mathbf{x}_t} D_{KL}\Big(q(\mathbf{x}_T|\mathbf{x}_t)\|p(\mathbf{x}_T|\mathbf{x}_t)\Big)}_{\mathbf{II}}
$$
(4)

As observed, the first control component (I) recovers the standard training-free diffusion guidance, where the posterior mean $\mathbf{x}_T$ is approximated via Tweedie's formula. To address the inherent limitations of such methods, we introduce an auxiliary control component (II) that enforces proximity between the proposal and true posteriors, ensuring $q(\mathbf{x}_T|\mathbf{x}_t) \approx p(\mathbf{x}_T|\mathbf{x}_t)$. However, since the true posterior has no closed-form expression, evaluating the KL term directly is infeasible. To address this, we adopt a particle-based optimization strategy using Stein variational inference. Given a set of $N$ particles $\mathcal{D}_t \leftarrow \left\{\mathbf{x}_t^i\right\}_{i=0}^{N}$ on the diffusion manifold $\mathcal{M}_t$, we evolve them along the KL most decreasing direction, which follows from Lemma 2.1:

$$
\phi^*(\mathbf{x}_t^i) = \mathbb{E}_{\mathbf{x}_T^j \sim q_{T|t}(\mathbf{x}_T|\mathbf{x}_t)}\Big[\nabla_{\mathbf{x}_t^j}\log p(\mathbf{x}_T^j|\mathbf{x}_t^j)k(\mathbf{x}_T^i, \mathbf{x}_T^j)
$$

$$
+\nabla_{\mathbf{x}_t^j}k(\mathbf{x}_T^i, \mathbf{x}_T^j)\Big]
$$
(5)

We initialize the proposal posterior $q_{T|t}(\mathbf{x}_T|\mathbf{x}_t)$ by a new set of particles via Tweedie's formula, i.e, $\mathcal{D}_T \leftarrow \Big\{\mathbf{x}_T^i|\mathbf{x}_T^i = \frac{\mathbf{x}_t^i + \gamma^2(t)\mathbf{s}_\theta(\mathbf{x}_t^i)}{\eta(t)}, \forall \mathbf{x}_t^i \in \mathcal{D}_t\Big\}$. However, directly computing this optimal control direction requires numerous Jacobian-vector products, which are memory-intensive in high-dimensional cases. To alleviate this memory burden, we propose a back-and-forth Stein correction.

**Back-and-Forth Stein Correction.** Figure 2 illustrates the key steps of the posterior correction mechanism, which first maps the noisy particles $\mathcal{D}_t$ *backward*, $\mathcal{M}_t \to \mathcal{M}_T$, to obtain the uncurated posterior samples $\mathcal{D}_T$, then applies the Stein correction on $\mathcal{D}_T$, and finally maps the corrected particles *forward* to the current diffusion manifold, $\mathcal{M}_T \to \mathcal{M}_t$. We follow the forward and backward (or reverse) conventions of Song et al. (2020b). The steepest descent direction for minimizing $-\nabla_{\mathbf{x}_T} D_{KL}\Big(q(\mathbf{x}_T|\mathbf{x}_t)\|p(\mathbf{x}_T|\mathbf{x}_t)\Big)$ on $\mathcal{M}_T$ can thus be computed more efficiently.

$$
\phi^*(\mathbf{x}_T^i) = \mathbb{E}_{\mathbf{x}_T^j \sim q_{T|t}(\mathbf{x}_T|\mathbf{x}_t)}\Big[\nabla_{\mathbf{x}_T^j}\log p(\mathbf{x}_T^j|\mathbf{x}_t^j)k(\mathbf{x}_T^i, \mathbf{x}_T^j)
$$

$$
+\nabla_{\mathbf{x}_T^j}k(\mathbf{x}_T^i, \mathbf{x}_T^j)\Big]
$$
(6)

As discussed, the true conditional posterior $p(\mathbf{x}_T|\mathbf{x}_t)$ has no closed-form expression; however, its score $\nabla_{\mathbf{x}_T}\log p(\mathbf{x}_T|\mathbf{x}_t)$ can be approximated using model scores, as established by the following result.

**Lemma 3.3.** *For* $\mathbf{x}_T \sim p(\mathbf{x}_T|\mathbf{x}_t)$*, the data conditional posterior score* $\nabla_{\mathbf{x}_T}\log p(\mathbf{x}_T|\mathbf{x}_t)$ *can be approximated in terms of the model scores* $\mathbf{s}_{\boldsymbol{\theta}}(\cdot)$ *as follows:*

$$
\nabla_{\mathbf{x}_T}\log p(\mathbf{x}_T|\mathbf{x}_t) \approx \mathbf{s}_\theta(\mathbf{x}_T) - \eta(t)\mathbf{s}_\theta(\mathbf{x}_t)
$$

Here, we again assume the forward kernel as $p_{t|T}(\mathbf{x}_t|\mathbf{x}_T) = \mathcal{N}(\eta(t)\mathbf{x}_T, \gamma^2(t)I)$. Applying this posterior score to Equation 6 yields the optimal evolving direction $\phi^*(\mathbf{x}_T^i)$ of the particles on $\mathcal{M}_T$. We now present the optimal controller for the surrogate stochastic optimal control.

**Proposition 3.4.** *Consider the low-density reward-based cost functional* $\widetilde{J}(\mathbf{u}, \mathbf{x}, t)$ *and its upper bound value function* $\bar{V}(\mathbf{x}, t, q)$*. Let* $q_{T|t}(\mathbf{x}_T|\mathbf{x}_t)$ *denote the proposal posterior initialized via Tweedie's formula, and let* $q_{T|t}^\epsilon(\mathbf{x}_T|\mathbf{x}_t)$ *denote the updated posterior obtained after applying the back-and-forth Stein correction with step size* $\epsilon(t)$*. Then, the optimal control* $\bar{\mathbf{u}}^*(\mathbf{x}, t)$ *for the surrogate value function* $\bar{V}(\mathbf{x}, t, q)$ *can be decomposed as:*

$$
\frac{\bar{\mathbf{u}}^*(\mathbf{x}_t^i, t)}{\sigma(t)} = \begin{matrix} -\alpha(t)\mathbf{s}_\theta(\mathbf{x}_t^i) \\ \underbrace{+\beta(t)\nabla_{\mathbf{x}_t^i}\mathbb{E}_{\mathbf{x}_T^i \sim q_{T|t}^\epsilon(\mathbf{x}_T|\mathbf{x}_t)}\Big[r(\mathbf{x}_T^i)\Big]}_{\textit{Low-density reward-based guidance on } \mathcal{M}_t} \end{matrix}
$$

$$
\oplus \underbrace{\mathbb{E}_{\mathbf{x}_T^j \sim q_{T|t}(\mathbf{x}_T|\mathbf{x}_t)}\Big[\Big(\mathbf{s}_\theta(\mathbf{x}_T^j) - \eta(t)\mathbf{s}_\theta(\mathbf{x}_t^j)\Big)k(\mathbf{x}_T^i, \mathbf{x}_T^j) \\ +\nabla_{\mathbf{x}_T^j}k(\mathbf{x}_T^i, \mathbf{x}_T^j)\Big]}_{\textit{Training-free Stein diffusion posterior correction on } \mathcal{M}_T}
$$

Here, $\oplus$ denotes the concatenation operator. The control factor associated with the Stein correction on $\mathcal{M}_T$ refines the initial proposal distribution $q_{T|t}(\mathbf{x}_T|\mathbf{x}_t)$ toward the true diffusion posterior, incurring the updated posterior $q_{T|t}^\epsilon(\mathbf{x}_T|\mathbf{x}_t) \approx p_{T|t}(\mathbf{x}_T|\mathbf{x}_t)$. After mapping the corrected particles forward to the noisy manifold $\mathcal{M}_t$, the second control factor guides the particles toward low-density regions with desired properties. Crucially, our Stein correction ensures robust and accurate guidance even when leveraging off-the-shelf classifiers on approximate posterior samples obtained via Tweedie's formula, thereby improving performance across diverse posterior sampling tasks. Figure 2

illustrates that standard training-free guidance methods often drift samples outside generative manifolds, whereas using Stein-corrected posteriors guarantees reliable guidance that keeps samples within them. We refer to the proposed method as Stein Diffusion Guidance (SDG), which we summarize in Algorithm 1 in the Appendix.

**Generalization of Langevin Correction.** SDG employs an adaptive step size $\epsilon(t)$ for particle updates, with its formulation provided in Appendix C.2. In the limit $\epsilon(t) \to 0$, the back-and-forth Stein correction can recover the Langevin correction from Song et al. (2020b).

**Corollary 3.5.** *Let the correction stepsize be set to zero, i.e., $\epsilon(t) = 0$ for all reverse steps $t$. Then, the back-and-forth Stein correction generalizes the Langevin correction as a special case with stepsize $\gamma^2(t)$ and noise scaled by $\sqrt{2}$:*

$$\mathbf{x}_t \leftarrow \mathbf{x}_t + \gamma^2(t)s_\theta(\mathbf{x}_t) + \gamma(t)\mathbf{z}, \qquad \mathbf{z} \sim \mathcal{N}(0, I)$$

Importantly, when $\epsilon(t) > 0$, SDG incorporates interactions between particles arising from repulsive forces, which play a nontrivial role in enhancing the guidance effectiveness toward desired targets. Similar forces have been used for non-i.i.d. diverse diffusion sampling (Corso et al., 2024).

# 4. Experiments

We first evaluate SDG on diverse image guidance tasks in non-low-density settings to assess its generalizability across different data domains (Section 4.1). We then perform more complex small-ligand sampling in low-density regions for solving protein-docking problems (Section 4.2). Depending on tasks, we ablate four different variants: SDG($\alpha(t) > 0, \epsilon(t) > 0$), SDG$^\clubsuit$(w/o Stein correction), SDG$^\heartsuit$($\alpha(t) = 0, \epsilon(t) > 0$), and SDG$^\diamond$($\alpha(t) > 0, \epsilon(t) = 0$).

## 4.1. Training-Free Diffusion Guidance on Image Tasks

We adapt the benchmarks from Ye et al. (2024) for four tasks: image label guidance, Gaussian deblurring, and super-resolution, text-to-image (T2I) style transfer. Detailed task descriptions, evaluation metrics, and baselines are provided in Appendix D. Since these tasks do not contain minority class samples, we disable low-density sampling ($\alpha(t) = 0$) and instead focus on generating samples with high desired properties. Table 1 indicates that SDG$^\heartsuit$ consistently outperforms its variant without the Stein correction SDG$^\clubsuit$, highlighting the importance of correcting Tweedie-based posterior samples before leveraging them for guidance. Furthermore, SDG$^\heartsuit$ surpasses relevant baselines, DPS (Chung et al., 2023) and LGD (Song et al., 2023a), which also rely exclusively on approximate posterior samples. Compared to more advanced baselines, UGD (Bansal et al., 2023) and MPGD (He et al., 2024), which employ recurrent strate-

gies, backward optimization, and manifold-preserving techniques, SDG$^\heartsuit$ still outperforms them on three out of four tasks. These results demonstrate that correcting approximate posterior samples is always crucial in training-free diffusion guidance tasks. Eventually, SDG can be readily integrated into any diffusion posterior sampling problem of interest without requiring model finetuning and/or distillation.

## 4.2. Low-Density Diffusion Guidance on Molecules

To evaluate the initial hypothesis that sampling molecules from low-density regions can improve drug discovery, we apply SDG to the ligand–protein docking problems adapted from Lee et al. (2023). In this setting, generated molecules or ligands must satisfy four hit conditions: (1) Tanimoto similarity (**SIM**) with the closest training sample in ZINC250k (Irwin et al., 2012) is below 0.4 to ensure novelty (**Nov.**); (2) synthetic accessibility score (**SA**) is below 5, indicating ease of molecular synthesis; (3) drug-likeness score (**QED**) exceeds 0.5; and (4) docking score (**DS**) is lower than the median DS of known actives. We evaluate this task on four protein targets: **Fa7** (Coagulation Factor VII), **5ht1b** (5-hydroxytryptamine receptor 1B), **Jak2** (Tyrosine-protein kinase JAK2), and **Parp1** (Poly[ADP-ribose] polymerase-1). Detailed task descriptions, evaluation metrics, and baselines are provided in Appendix C.

**Sampling Novel Hit Molecules.** Table 2 shows that, without the Stein correction, SDG$^\clubsuit$ performs poorly, indicating that Tweedie's formula alone provides biased and unreliable approximate diffusion posteriors. Samples from these posteriors fail to offer reliable guidance, particularly in low-density regions where score-based models are least accurate due to limited training data support. In contrast, SDG substantially enhances guidance by leveraging Stein-corrected posterior samples, leading to improvements of several orders of magnitude. This not only validates the theoretical motivation for the Stein correction but also demonstrates consistent empirical gains. Moreover, SDG also outperforms the baselines, the pretrained model GDSS (Jo et al., 2022), and its classifier-guidance variant MOOD (Lee et al., 2023), on two protein targets. Compared to non-diffusion methods, SDG generates more hit compounds across the target proteins. Figure 5 compares the distributions of docking scores. Without the Stein correction, SDG$^\clubsuit$ fails to align with the data distributions, resulting in poor guidance and fewer promising candidates. In contrast, SDG effectively regularizes sampling in low-density regions, shifting the docking score distributions toward the desired range and enabling the generation of more potential hit compounds.

Table 2 reveals that optimizing solely for rewards ($\alpha(t) = 0$) leads SDG$^\heartsuit$ to perform sub-optimally, generating fewer novel hit molecules compared to the full SDG setting ($\alpha(t) > 0, \epsilon(t) > 0$). This result further underscores the

*Table 1.* Comparison of training-free diffusion guidance methods on *non–low-density* image guidance tasks. Baseline results and evaluation metrics are taken from Ye et al. (2024). The most relevant metrics are shown in bold, and the best results are highlighted. † SDG variants can be readily integrated into the Ye et al. (2024) framework as a plug-and-play module (see Appendix D for further details).

| Method | LABEL GUIDANCE | | GAUSSIAN DEBLUR | | SUPER RESOLUTION | | T2I STYLE TRASFER† | |
| --- | --- | --- | --- | --- | --- | --- | --- | --- |
| | **Accuracy (%)** ↑ | FID ↓ | **LPIPS** ↓ | FID ↓ | **LPIPS** ↓ | FID ↓ | **Style Score** ↓ | CLIP Score ↑ |
| DPS | 50.1 | 172.0 | 0.390 | 98.30 | 0.420 | 109.0 | 5.06 | 31.7 |
| LGD | 32.2 | 102 | 0.270 | 85.1 | 0.360 | 96.7 | 5.42 | 31.3 |
| MPGD | 38.0 | 88.3 | 0.177 | 69.3 | 0.283 | 82.0 | 4.08 | 31.5 |
| UGD | 45.9 | 94.2 | 0.200 | 69.3 | 0.249 | 75.9 | 4.97 | 31.5 |
| SDG♣ | 48.2 | 89.50 | 0.326 | 87.23 | 0.315 | 91.99 | 3.16 | 29.0 |
| SDG♡ | 54.0 | 105.4 | 0.246 | 70.00 | 0.228 | 68.90 | 3.05 | 28.8 |

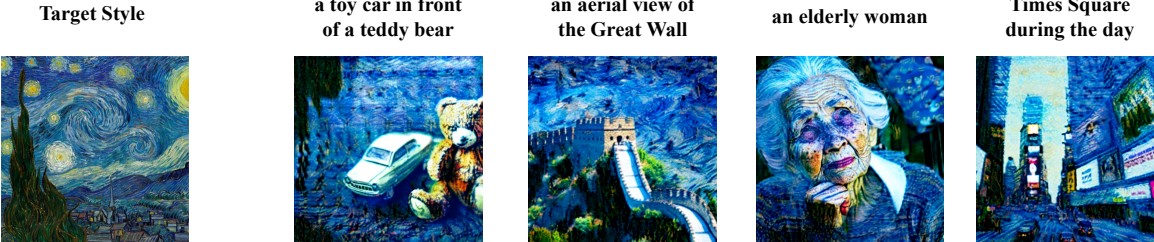

| Target Style | a toy car in front of a teddy bear | an aerial view of the Great Wall | an elderly woman | Times Square during the day |

*Figure 3.* Generated examples from the text-to-image (T2I) style transfer task with SDG♡. The target-style image is taken from WikiArt (Saleh & Elgammal, 2015), and the four prompts are sampled from Partiprompts (Yu et al., 2022).

*Table 2.* Mean and standard deviation of novel hit ratio (%) over three runs; baseline results from Lee et al. (2023). $\alpha(t)$ and $\epsilon(t)$ control low-density levels and particle update rates, respectively.

| Method | NOVEL HIT RATIO (%) ↑ | | | |
| --- | --- | --- | --- | --- |
| | Fa7 | 5ht1b | Jak2 | Parp1 |
| HierVAE | 0.007 (±0.013) | 0.507 (± 0.278) | 0.227 (±0.127) | 0.553 (±0.214) |
| MORLD | 0.007 (±0.013) | 0.880 (± 0.735) | 0.227 (±0.118) | 0.047 (±0.050) |
| FREED | 1.107 (±0.209) | 10.187 (± 3.306) | 4.520 (±0.673) | 3.627 (±0.961) |
| GDSS | 0.368 (±0.103) | 4.667 (± 0.306) | 1.167 (±0.281) | 1.933 (±0.208) |
| MOOD | 0.733 (±0.141) | 18.673 (±0.423) | 9.200 (±0.524) | 7.017 (±0.428) |
| SDG | 1.156 (±0.087) | 22.690 (±0.341) | 9.167 (±0.262) | 8.780 (±1.033) |
| SDG♣ | 0.299 (±0.094) | 0.033 (±0.027) | 0.000 (±0.000) | 0.671 (±0.302) |
| SDG♡ | 0.915 (±0.031) | 21.278 (±0.332) | 8.312 (±0.541) | 8.300 (±0.067) |
| SDG♢ | 0.956 (±0.247) | 21.722 (±0.275) | 8.722 (±0.218) | 7.933 (±0.233) |

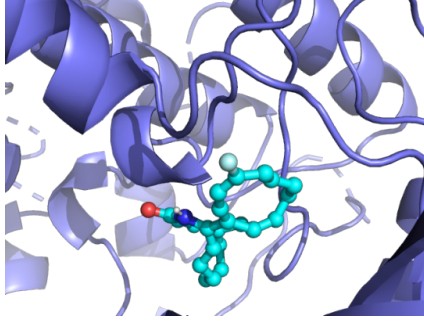

sim=0.30, ds=-11.00, qed=0.77, sâ=0.76

*Figure 4.* Example of the docking pose and optimized property values of a sampled ligand with SDG bound to the Jak2 receptor.

importance of explicitly targeting low-density regions in molecular sampling tasks (refer to Appendix C.3.5 for extreme low-density sampling ablations). Moreover, SDG incorporates a novel Stein correction, which is empirically more robust than the Langevin variant SDG♢ of Corollary 3.5. These performance gains stem from the non-trivial interaction forces between particles, previously leveraged to increase sample diversity. Unlike Corso et al. (2024), however, SDG does not suffer from diversity issues, as most generated molecules remain unique (Figure 7). Figure 4 visualizes a sampled ligand docked on the Jak2 protein.

**Reward Overestimation and Off-Manifold Sampling.**
In many applications, true (genuine) rewards are computed by non-differentiable oracle functions, which cannot be directly used in training-free diffusion guidance methods. Reward models and classifiers are trained to learn these genuine rewards and produce approximate (nominal) rewards, serving as differentiable proxies for diffusion models. However, due to the finite number of training samples, reward models tend to provide reliable signals only within the training data support. This limitation becomes more severe in low-density sampling problems. Uehara et al. (2024) first formulated this issue and proposed an entropy-regularized control approach, which is equivalent to solving a costly stochastic optimal control problem. In our case study, the absence of regularization leads the reward models to produce highly unreliable estimates of the true rewards during the sampling process (Figure 6.b), termed as *reward over-*

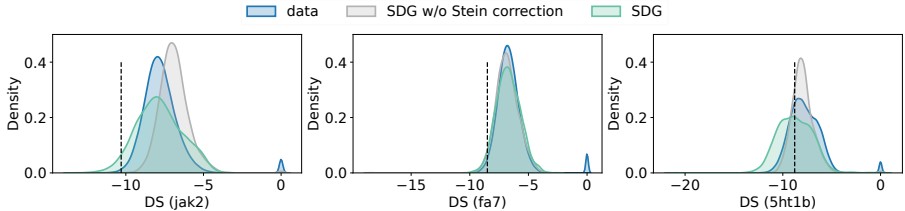

*Figure 5.* Distribution of docking scores (lower is better) for generated molecules with, SDG, and without Stein correction, SDG♣. The black dashed lines indicate thresholds for identifying hit compounds.

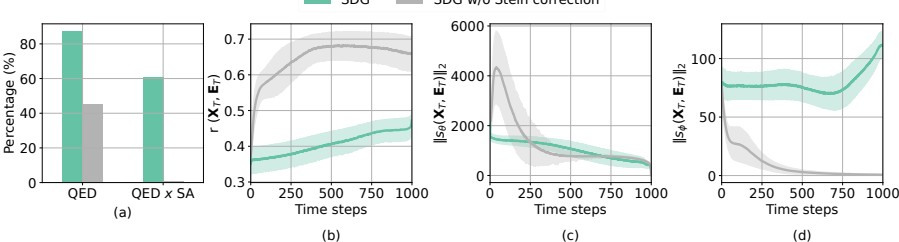

*Figure 6.* Temporal sampling dynamics of SDG for the Jak2 protein. (a) Percentage of molecules meeting QED and SA hit criteria; (b) Rewards of posterior samples $r(\mathbf{X}_T, \mathbf{E}_T)$; (c) Frobenius norm of node posterior scores $s_\theta(\mathbf{X}_T, \mathbf{E}_T)$; (d) Frobenius norm of edge posterior scores $s_\phi(\mathbf{X}_T, \mathbf{E}_T)$.

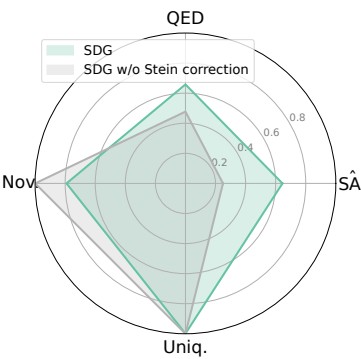

*Figure 7.* Multiple sampling objectives on Jak2, including sample novelty (Nov.), drug-likeness (QED), normalized synthetic accessibility (SÂ), and sample uniqueness (Uniq.). Higher scores indicate better performance.

*estimation.* Moreover, this leads to the generation of novel molecular structures that lack drug-likeness and are difficult to synthesize (Figures 6.a and 7). Additionally, the model scores vanish and are misdirected away from generative manifolds (Figure 6.c and 6.d). Further details on the sampling score dynamics are provided in Appendix C.3.3. In contrast, the Stein correction enables SDG to act as an efficient regularization mechanism that (i) improves genuine reward estimation accuracy (Figure 6.b), (ii) promotes the generation of more realistic and synthesizable molecules (Figure 6.a and Figure 7), and (iii) preserves sampling within the generative manifold (Figures 6.c and 6.d), particularly under challenging low-density conditions.

**Effect of Particle Size on Stein Correction.** SDG relies on Stein variational inference, whose effectiveness depends on the number of particles. While theory guarantees better

*Table 3.* Ablation results on the number of particles.

| Receptor | NOVEL HIT RATIO (%) ↑ | | |
|---|---|---|---|
| | 512 | 1024 | 3000 |
| Fa7 | $1.156_{(\pm 0.087)}$ | $1.044_{(\pm 0.063)}$ | $1.067_{(\pm 0.109)}$ |
| 5ht1b | $22.690_{(\pm 0.341)}$ | $21.922_{(\pm 0.461)}$ | $22.389_{(\pm 0.490)}$ |
| Jak2 | $9.078_{(\pm 0.278)}$ | $8.822_{(\pm 0.532)}$ | $9.167_{(\pm 0.262)}$ |

approximation with more particles (Liu & Wang, 2016), our empirical results reveal a saturation point beyond which additional particles can yield inconsistent performance gains. Table 3 presents an ablation study across varying particle sizes, showing that larger particle sizes do not necessarily lead to better results. These behaviors likely arise from the inherent instability of kernel-based updates in high-dimensional spaces under the standard SVGD framework (Liu & Wang, 2016), consistent with prior analyses in Zhang et al. (2020), and highlight the need for more robust particle update schemes.

### 4.3. Computational Analysis

*Table 4.* Computation efficiency analysis of the back-and-forth Stein correction on the image-label (Im.) and molecule (Mol.) guidance tasks. Detailed setup is referred to Appendices C.3.6, D.

| Task | Guidance | Runtime (s) | Memory (MB) |
|---|---|---|---|
| Mol. | SDG♣ | 1360 | 7783 |
| | SDG | $2521_{(\uparrow+85.4\%)}$ | $8049_{(\uparrow+3.4\%)}$ |
| Im. | SDG♣ | 602 | 18728 |
| | SDG | $820_{(\uparrow+36.2\%)}$ | $18792_{(\uparrow+0.3\%)}$ |

Table 4 compares the computational resources consumed by SDG variants. We observe that SDG incurs only a small memory overhead compared to SDG♣ (without Stein correction), highlighting the effectiveness of the proposed back-and-forth mechanism. In addition, despite the $\mathcal{O}(n^2)$ memory complexity of Stein updates with respect to the number of particles $n$, SDG still scales to large particle sizes ($n = 3000$) for molecule guidance without imposing a significant memory burden in practice. This is because the memory cost of pairwise particle interactions is typically negligible compared to the memory required for score-model evaluations (Liu & Wang, 2016). Furthermore, SDG demonstrates strong scalability across different ambient data dimensions and tasks. In terms of speed, SDG nearly doubles the running time for molecule guidance when using the standard SDE solvers of Jo et al. (2022). However, with the faster DDIM solver (Song et al., 2020a), SDG reduces the runtime overhead to approximately $36\%$, emphasizing the importance of deploying SDG with more efficient solvers.

## 5. Related Work

**Diffusion Models.** The idea of learning to reverse a diffusion process, introduced by Sohl-Dickstein et al. (2015), has led to the development of several advanced methods (Ho et al., 2020; Song et al., 2020b;a). Later work by Dhariwal & Nichol (2021) incorporated classifier guidance to steer generated samples toward regions with desired properties. Ho & Salimans (2022) further proposed classifier-free guidance, which removes the need for an explicit classifier while achieving a similar objective. However, classifier-free guidance typically requires training conditional and unconditional models jointly, which adds complexity to the training process. More recently, Sadat et al. (2025) reduced this complexity by perturbing time and/or label conditions.

**Diffusion Posterior Sampling.** In contrast to methods that guide generation using noisy intermediate states $x_t$ (Dhariwal & Nichol, 2021; Skreta et al., 2025), diffusion posterior sampling (Chung et al., 2023) performs guidance using posterior samples $x_T$. This enables the use of off-the-shelf classifiers (Ye et al., 2024) and has been applied to a wide range of problems (Chung et al., 2023; MOUFAD et al., 2025; Yu et al., 2023; Ma et al., 2024; Shen et al., 2024; Rout et al., 2025; Janati et al., 2025; Dinh et al., 2025).

**Stochastic Optimal Control.** Stochastic optimal control aims to control a stochastic dynamical system so as to minimize a user-defined cost function. Its core theoretical foundation is the verification theorem (Fleming & Soner, 2006; Pham, 2009; Nüsken & Richter, 2021). The pioneering work of Uehara et al. (2024) first generalized this framework to diffusion guidance settings, but its practical applications remain computationally prohibitive.

**Stein Variational Inference.** Liu & Wang (2016) introduced Stein variational gradient descent, a non-parametric method that approximates a target distribution by transporting particles. Subsequent work improved its convergence by incorporating geometric preconditioning (Wang et al., 2019) and by formulating the method as a Wasserstein gradient flow of the chi-squared divergence (Chewi et al., 2020).

## 6. Conclusion

We propose Stein Diffusion Guidance, a low-cost alternative to SOC's methods for enhancing diffusion guidance in a training-free manner. By analyzing the existing biases of Tweedie-based approximate posteriors through the lens of SOC theory, we introduce a plug-and-play Stein correction that effectively mitigates these biases. Experiments on low-density molecular sampling and image guidance tasks provide strong empirical support for our theoretical claims.

**Limitations & Perspectives.** While effective, SDG still inherits the limitations of standard SVGD frameworks. Notably, SDG derives the KL descent direction that is steepest within the class of RKHS transport perturbations (Lemma 2.1). However, this RKHS-restricted approximation may become less stable or less effective in high-dimensional settings. Future work could explore more stable SVGD variants, including geometry-informed frameworks (Wang et al., 2019; Chewi et al., 2020), to further improve SDG for low-density diffusion guidance in high-dimensional settings. In addition, due to the back-and-forth Stein correction mechanism, SDG typically incurs a runtime overhead compared to standard training-free guidance baselines. To accelerate the sampling speed of SDG for fast-inference applications, one could also combine SDG with few-step consistency diffusion models (Song et al., 2023b).

## Acknowledgments

We acknowledge financial support from the Swiss National Science Foundation through the LegoMol project (grant no. 207428). The computations were partially performed on the Baobab cluster at the University of Geneva. We also thank the anonymous ICML reviewers for their insightful feedback and suggestions.

## Impact Statement

SDG represents a novel training-free diffusion guidance approach for low-density ligand sampling in docking problems, with the potential to advance drug discovery by identifying effective drug candidates for cancer treatment. However, in the wrong hands, it could be misused to design harmful or addictive substances illicitly. Ensuring responsible use is therefore critical to maximizing its positive societal impact.

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

# A. Additional Qualitative Results

## A.1. Visualization Results on Image Diffusion Guidance Tasks

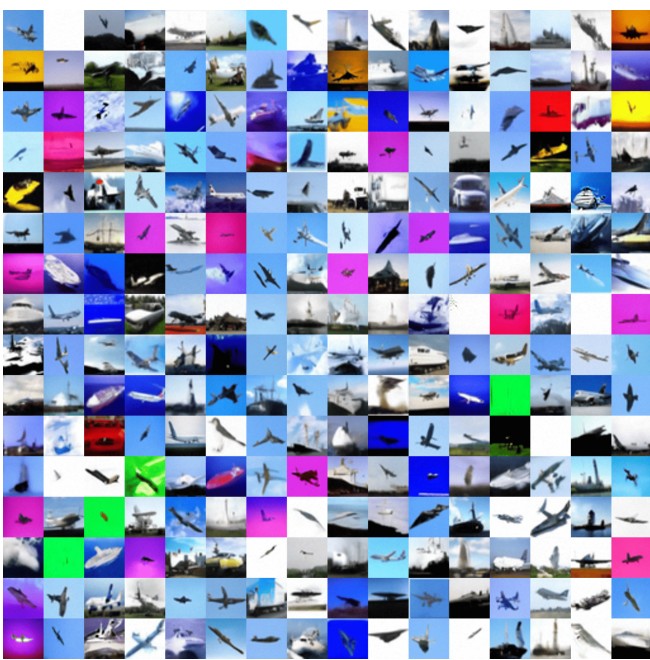

*Figure 8.* Image label guidance results without the Stein correction SDG♣ (CIFAR10 class: Airplane).

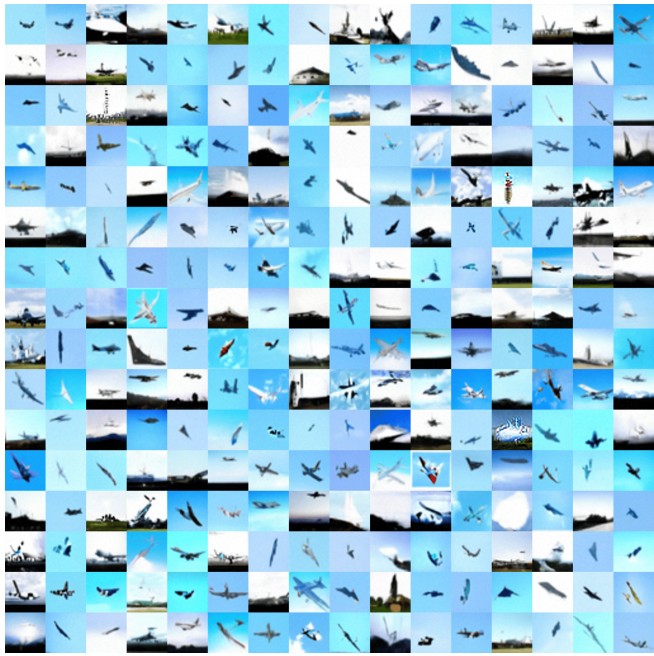

*Figure 9.* Image label guidance results with the Stein correction SDG♡ (CIFAR10 class: Airplane).

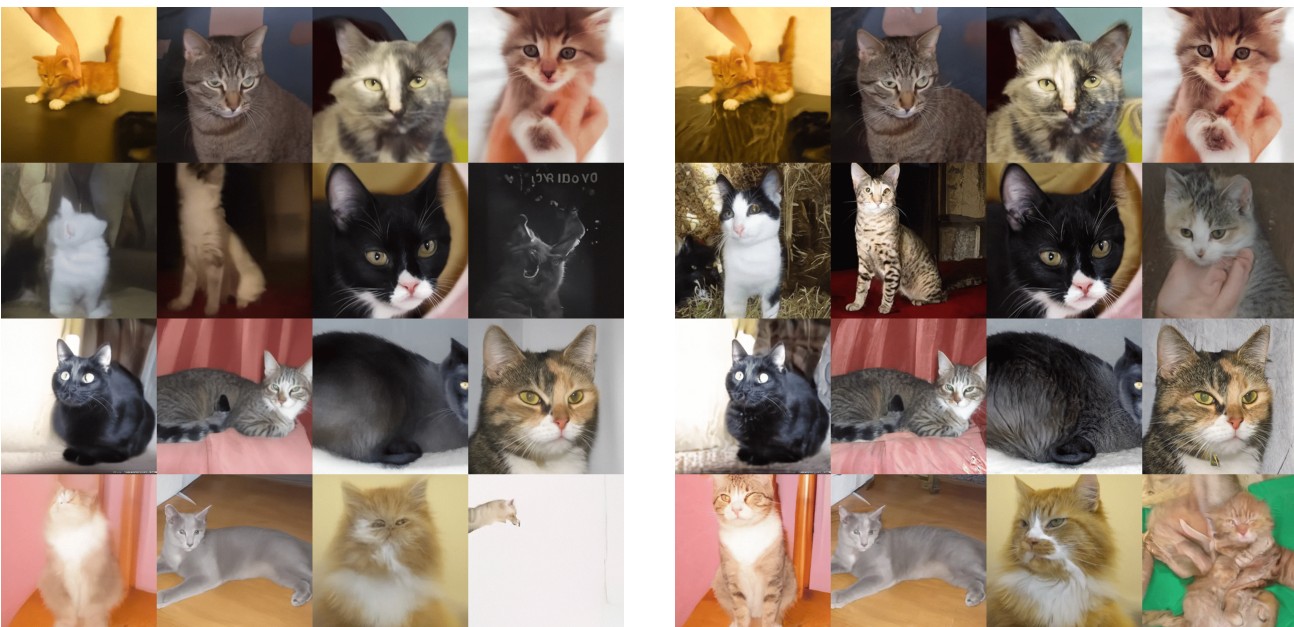

*Figure 10.* Visualization of image deblurring results without, SDG♣ (Left), versus with, SDG♡ (Right), the Stein correction.

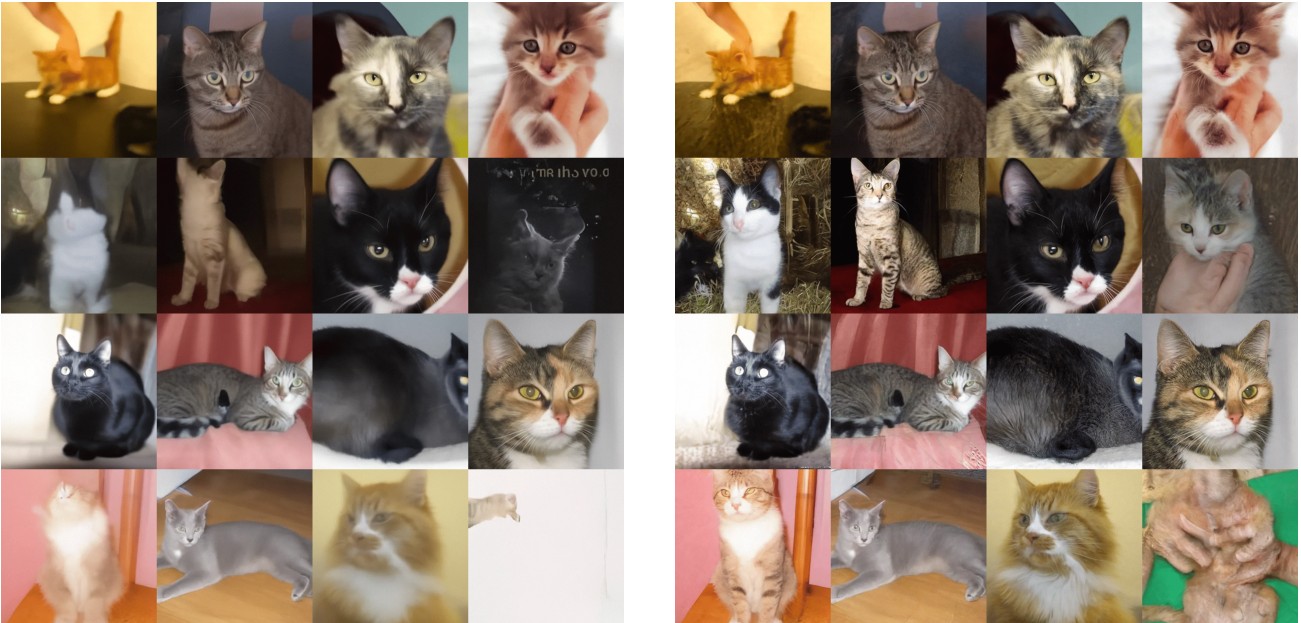

*Figure 11.* Visualization of image super-resolution results without, SDG♣ (Left), versus with, SDG♡ (Right), the Stein correction.

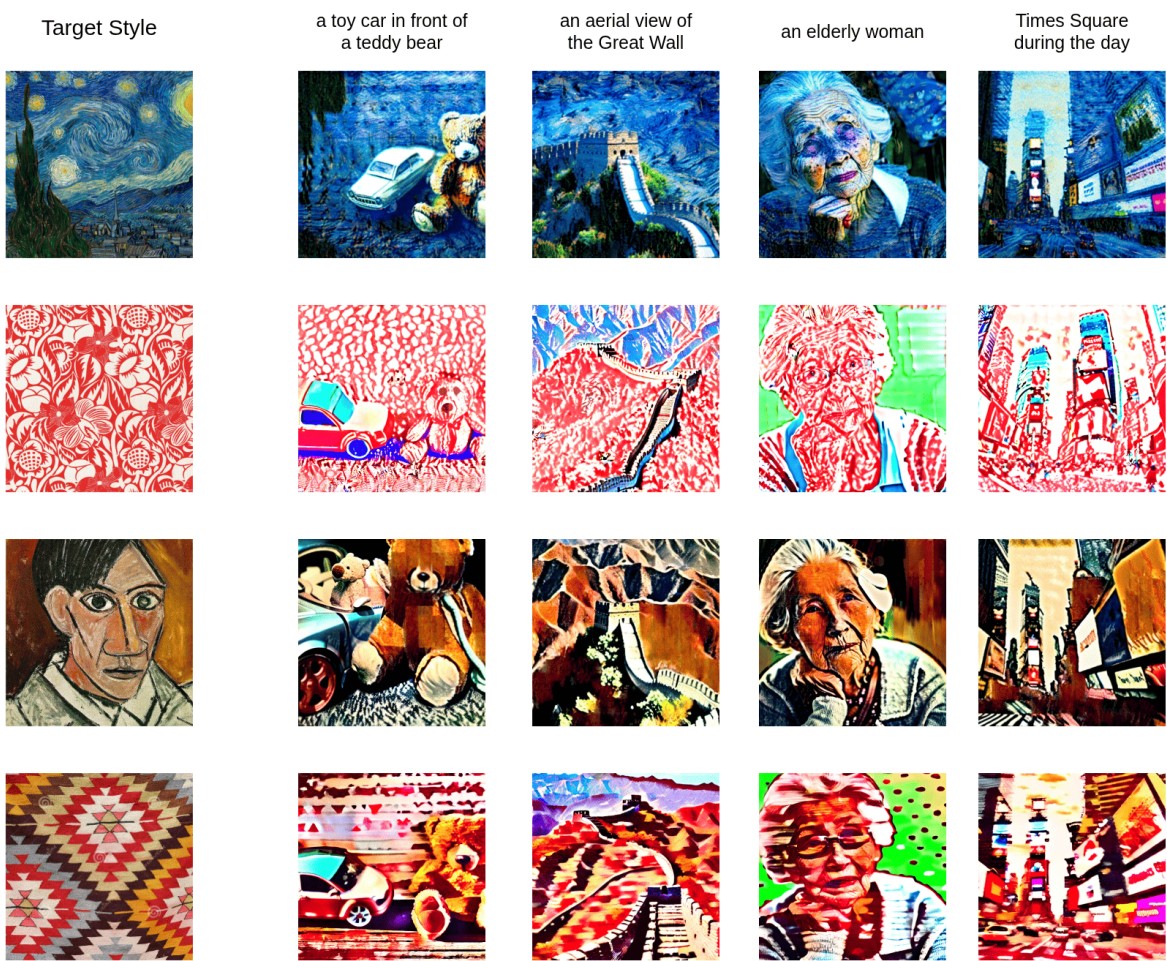

*Figure 12.* Text-to-image style transfer generated samples with SDG$^\heartsuit$ .

## A.2. Visualization Results on Ligand-Protein Docking Tasks

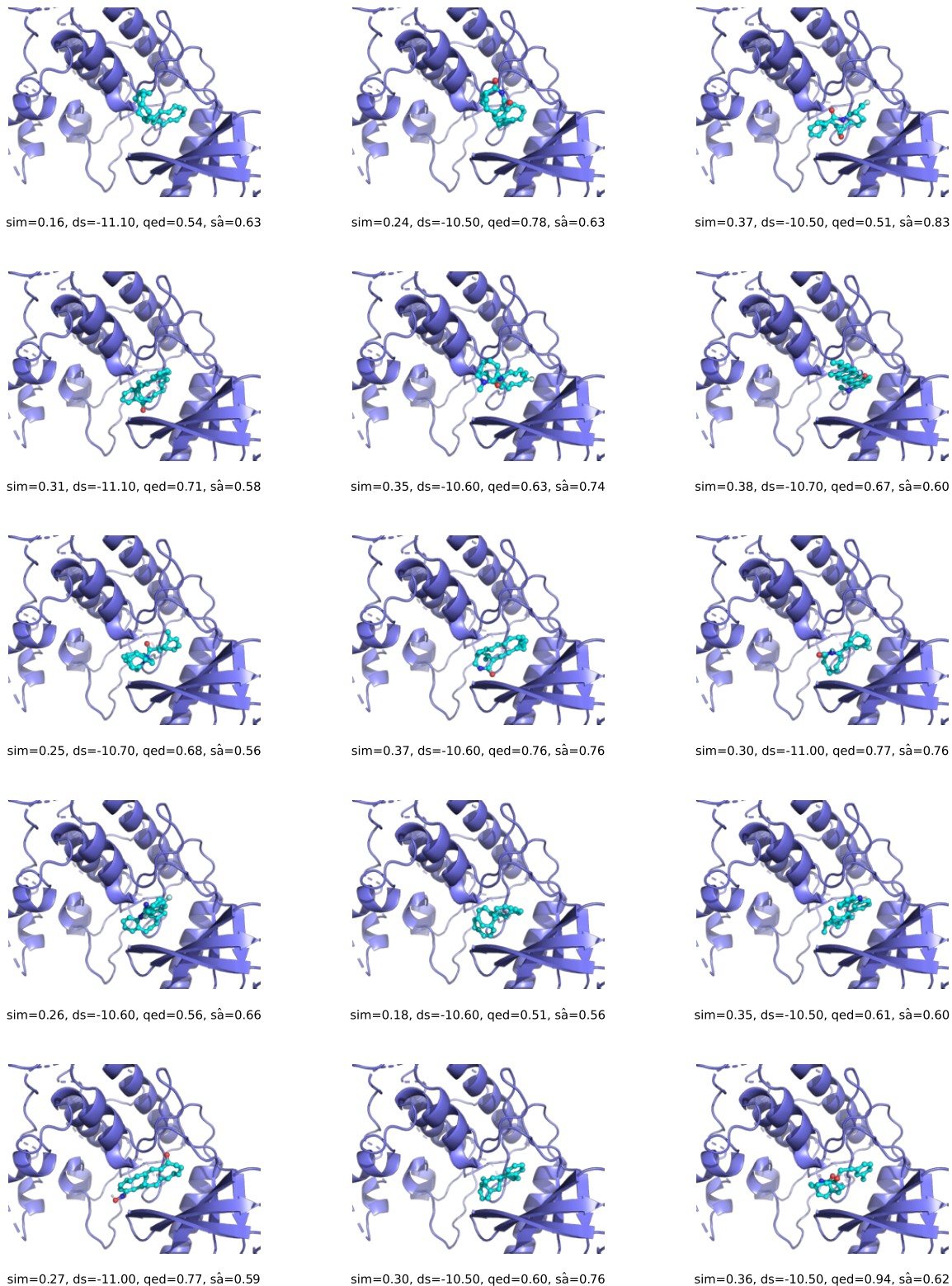

*Figure 13.* Visualization of docking poses for multiple generated ligands from SDG bound to the Jak2 protein.

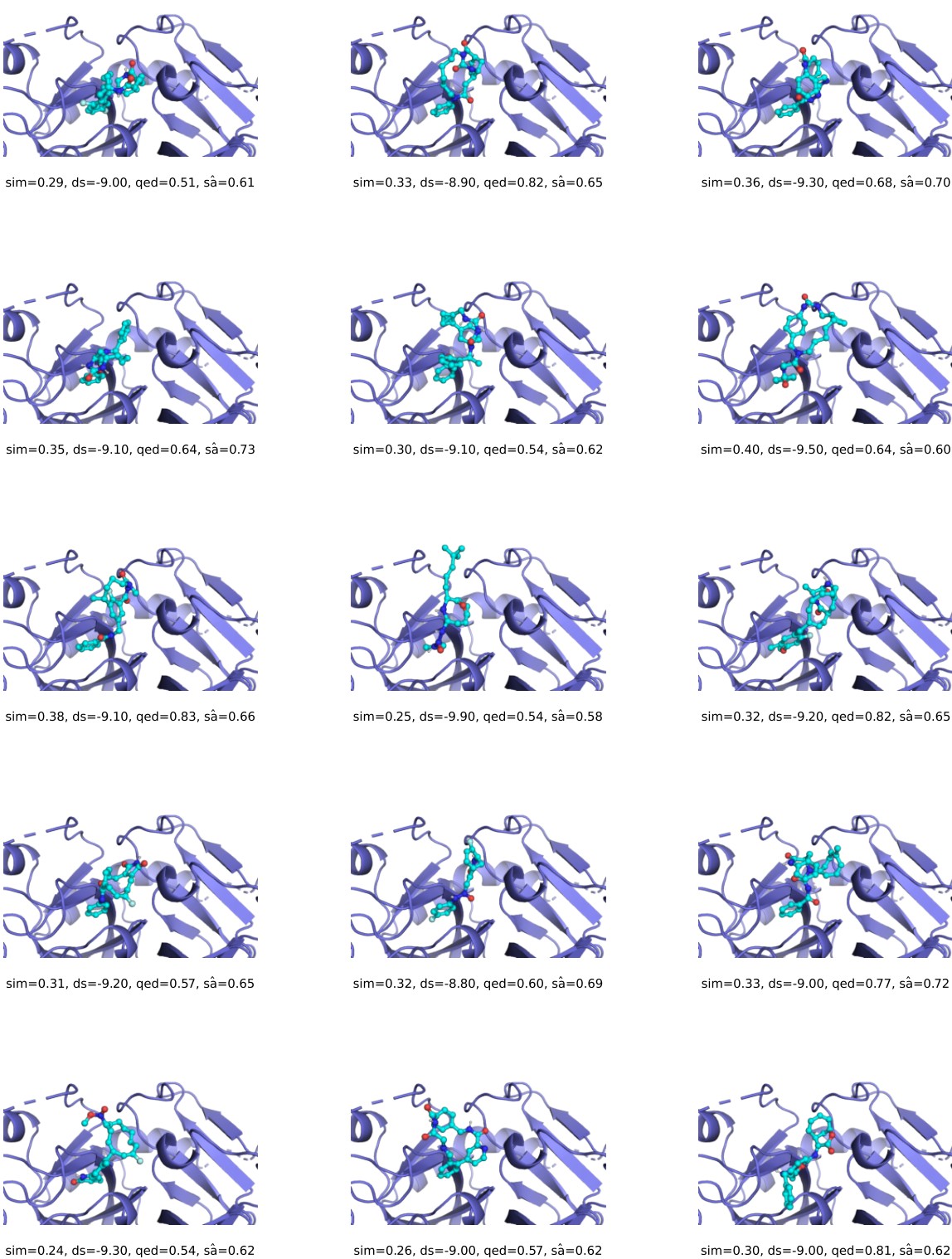

*Figure 14.* Visualization of docking poses for multiple generated ligands from SDG bound to the Fa7 protein.

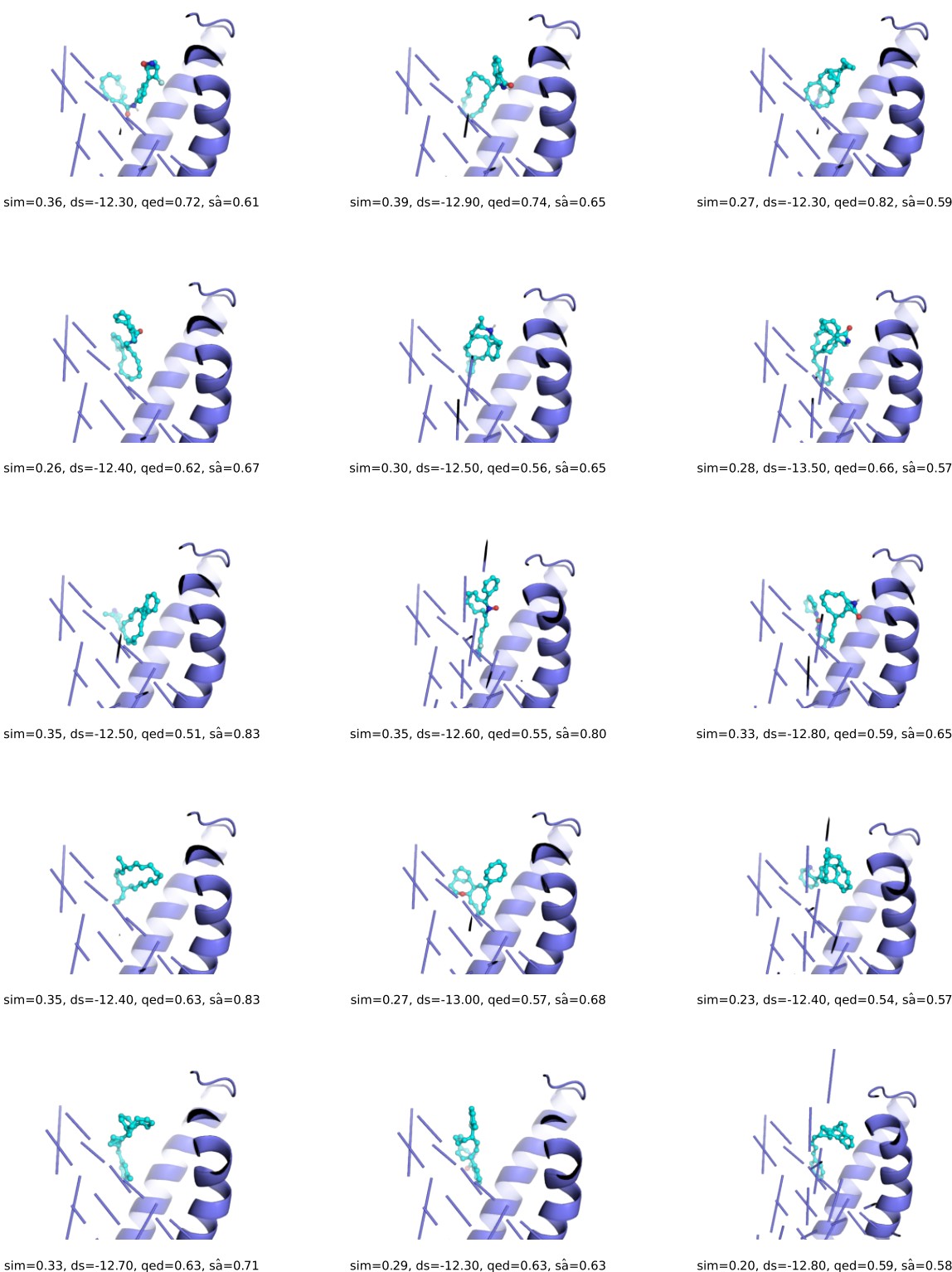

*Figure 15.* Visualization of docking poses for multiple generated ligands from SDG bound to the 5ht1b protein.

---

**Algorithm 1** Stein Diffusion Guidance Algorithm

---

1: **Input**: Score model $s_\theta(\mathbf{x}_t)$, number of particles $N$, off-the-shelf classifier $r(\mathbf{x}_T)$, correction step size $\epsilon(t)$, low-density schedule $\alpha(t)$, guidance strength schedule $\beta(t)$, total steps $T$.

2: **Output**: Endpoint samples.

$\quad \mathcal{D}_0 \leftarrow \left\{ \mathbf{x}_0^i | \mathbf{x}_0^i \sim p_0, 1 \leqslant i \leqslant N \right\}$ {initial particles}

3: **for** $t$ in $[0, T)$:

$\quad$ /* Back-and-forth Stein correction */

4: $\quad \mathcal{D}_T \leftarrow \left\{ \mathbf{x}_T^i | \mathbf{x}_T^i = \frac{\mathbf{x}_t^i + \gamma^2(t) s_\theta(\mathbf{x}_t^i)}{\eta(t)}, \forall \mathbf{x}_t^i \in \mathcal{D}_t \right\}$ {Backward mapping}

5: $\quad \mathbf{x}_T^i = \mathbf{x}_T^i + \epsilon(t) \frac{1}{N} \sum_{\mathbf{x}_t^j, \mathbf{x}_T^j \in \mathcal{D}_t \bigcup \mathcal{D}_T} \left( (s_\theta(\mathbf{x}_T^j) - \eta(t) s_\theta(\mathbf{x}_t^j)) k(\mathbf{x}_T^i, \mathbf{x}_T^j) + \nabla_{\mathbf{x}_T^j} k(\mathbf{x}_T^i, \mathbf{x}_T^j) \right)$

6: $\quad \mathcal{D}_t \leftarrow \left\{ \mathbf{x}_t^i | \mathbf{x}_t^i \sim \mathcal{N}(\eta(t) \mathbf{x}_T^i, \gamma^2(t) I), \forall \mathbf{x}_T^i \in \mathcal{D}_T \right\}$ {Forward mapping}

$\quad$ /* Low-density reward-based diffusion guidance */

7: $\quad \mathbf{x}_T^i = \frac{\mathbf{x}_t^i + \gamma^2(t) s_\theta(\mathbf{x}_t^i)}{\eta(t)}$

8: $\quad \mathbf{x}_{t+1}^i = \mathbf{x}_t^i - \mathbf{b}(\mathbf{x}_t^i, t) + \sigma(t)^2 \left( (1 - \alpha(t)) s_\theta(\mathbf{x}_t^i) + \beta(t) \nabla_{\mathbf{x}_t^i} r(\mathbf{x}_T^i) \right) + \sigma(t) \mathbf{z}, \quad \mathbf{z} \sim \mathcal{N}(0, I)$

9: **return**: $\left\{ \mathbf{x}_T^i, 1 \leqslant i \leqslant N \right\}$

---

## B. Theoretical Proofs

**Proposition 3.1** *Consider the stochastic optimal control problem with the novel functional cost $\widetilde{J}(\mathbf{u}, \mathbf{x}, t)$ defined in Equation 3. By Lemma 2.2, the marginal density $p_t^{\mathbf{u}}(\mathbf{x}_t)$, the value function $V(\mathbf{x}, t)$, and the optimal control $\mathbf{u}^*(\mathbf{x}, t)$ of the controlled-reverse SDE under $\mathbb{P}^{\mathbf{u}}$ are computed as:*

$$p_t^{\mathbf{u}}(\mathbf{x}_t) = \mathbb{E}_{\mathbb{P}} \left[ p_t^{1-\alpha(t)}(\mathbf{x}_t) \exp(\beta(t) r(\mathbf{x}_T)) | \mathbf{x}_t \right]$$

$$\mathbf{u}^*(\mathbf{x}, t) = \sigma(t) \nabla_{\mathbf{x}} \log \frac{p_t^{\mathbf{u}}(\mathbf{x})}{p_t(\mathbf{x})} \quad \text{and} \quad V(\mathbf{x}, t) = -\log \frac{p_t^{\mathbf{u}}(\mathbf{x})}{p_t(\mathbf{x})}$$

*Proof.* For low-density sampling, we introduce a state cost that penalizes the density of the current state, expressed as $f(\mathbf{x}_s, s) = \alpha(s) \log p_s(\mathbf{x}_s) \delta(s - t)$, where $\alpha(s)$ defines a low-density annealing schedule. To maximize the reward of generated samples, we define a terminal cost as $g(\mathbf{x}_T) = -\beta(t) r(\mathbf{x}_T)$, where $r(\cdot)$ is the reward function and $\beta(t)$ defines a guidance-strength schedule. These two terms form a novel functional cost function $\widetilde{J}(\mathbf{u}, \mathbf{x}, t)$ (Equation 3 ). Since both the density term $p_s(\mathbf{x}_s)$ (associated with the state cost $f(\mathbf{x}_s, s)$) and the reward function $r(\mathbf{x}_T)$ (associated with the terminal cost $g(\mathbf{x}_T)$) possess continuous first-order spatial derivatives, as required by Lemma 2.2, we obtain the value function for this cost function.

$$
\begin{aligned}
V(\mathbf{x}, t) &= -\log \mathbb{E}_{\mathbb{P}} \left[ \exp \left( -\int_t^T \alpha(s) \log p_s(\mathbf{x}_s) \delta(s - t) ds + \beta(t) r(\mathbf{x}_T) \right) | \mathbf{x}_t = \mathbf{x} \right] \\
&= -\log \mathbb{E}_{\mathbb{P}} \left[ \exp \left( -\alpha(t) \log p_t(\mathbf{x}) + \beta(t) r(\mathbf{x}_T) \right) | \mathbf{x}_t = \mathbf{x} \right] \\
&= -\log \mathbb{E}_{\mathbb{P}} \left[ p_t^{-\alpha(t)}(\mathbf{x}) \exp \left( \beta(t) r(\mathbf{x}_T) \right) | \mathbf{x}_t = \mathbf{x} \right]
\end{aligned}
\tag{7}
$$

And, the optimal control has the form.

$$\mathbf{u}^*(\mathbf{x}, t) = \sigma(t) \nabla_{\mathbf{x}} \log \mathbb{E}_{\mathbb{P}} \left[ p_t^{-\alpha(t)}(\mathbf{x}) \exp \left( \beta(t) r(\mathbf{x}_T) \right) | \mathbf{x}_t = \mathbf{x} \right] \tag{8}$$

Substituting the optimal control into the stochastic optimal control problem (Equation 2) yields the controlled SDE under $\mathbb{P}^{\mathbf{u}}$.

$$
\begin{aligned}
d\mathbf{x}_t^{\mathbf{u}} &= \left(-\mathbf{b}(\mathbf{x}_t^{\mathbf{u}}, t) + \sigma(t)^2 \nabla_{\mathbf{x}_t^{\mathbf{u}}} \log p_t(\mathbf{x}_t^{\mathbf{u}}) + \sigma(t)\mathbf{u}\left(\mathbf{x}_t^{\mathbf{u}}, t\right)\right)dt + \sigma(t)d\mathbf{w} \\
&= \left(-\mathbf{b}(\mathbf{x}_t^{\mathbf{u}}, t) + \sigma(t)^2 \nabla_{\mathbf{x}_t^{\mathbf{u}}} \log p_t(\mathbf{x}_t^{\mathbf{u}}) + \sigma(t)^2 \nabla_{\mathbf{x}} \log \mathbb{E}_{\mathbb{P}}\left[p_t^{-\alpha(t)}\left(\mathbf{x}\right)\exp\left(\beta(t)r(\mathbf{x}_T)\right)|\mathbf{x}_t^{\mathbf{u}} = \mathbf{x}\right]\right)dt + \sigma(t)d\mathbf{w} \\
&= \left(-\mathbf{b}(\mathbf{x}_t^{\mathbf{u}}, t) + \sigma(t)^2 \nabla_{\mathbf{x}} \log \mathbb{E}_{\mathbb{P}}\left[p_t^{1-\alpha(t)}(\mathbf{x})\exp\left(\beta(t)r(\mathbf{x}_T)\right)|\mathbf{x}_t^{\mathbf{u}} = \mathbf{x}\right]\right)dt + \sigma(t)d\mathbf{w} \\
&= \left(-\mathbf{b}(\mathbf{x}_t^{\mathbf{u}}, t) + \sigma(t)^2 \nabla_{\mathbf{x}} \log p_t^{\mathbf{u}}(\mathbf{x})\right)dt + \sigma(t)d\mathbf{w}
\end{aligned}
\tag{9}
$$

where $p_t^{\mathbf{u}}(\mathbf{x}_t)$ denotes the annealed, reward-guided marginal density under $\mathbb{P}^{\mathbf{u}}$:

$$
p_t^{\mathbf{u}}(\mathbf{x}_t) = \mathbb{E}_{\mathbb{P}}\left[p_t^{1-\alpha(t)}(\mathbf{x}_t)\exp(\beta(t)r(\mathbf{x}_T))|\mathbf{x}_t\right]
\tag{10}
$$

Substituting this expression back into the value function and optimal control, we conclude the proof.

$$
\mathbf{u}^*(\mathbf{x}, t) = \sigma(t)\nabla_{\mathbf{x}} \log \frac{p_t^{\mathbf{u}}(\mathbf{x})}{p_t(\mathbf{x})} \quad \text{and} \quad V(\mathbf{x}, t) = -\log \frac{p_t^{\mathbf{u}}(\mathbf{x})}{p_t(\mathbf{x})}
\tag{11}
$$

**Proposition 3.2** *Consider a proposal distribution $q$ from a traceable family of distributions $\mathcal{Q}$. Then, the value function in Proposition 3.1 admits the following variational bound:*

$$
\begin{aligned}
V(\mathbf{x}, t) &\leq \bar{V}(\mathbf{x}, t, q) \\
&= -\mathbb{E}_{\mathbf{x}_T \sim q_{T|t}(\mathbf{x}_T|\mathbf{x})}\left[\log\left(p_t^{-\alpha(t)}(\mathbf{x})\exp\left(\beta(t)r(\mathbf{x}_T)\right)\right)|\mathbf{x}_t = \mathbf{x}\right] + D_{KL}\left(q(\mathbf{x}_T|\mathbf{x}_t)\|p(\mathbf{x}_T|\mathbf{x}_t)\right)\Big|_{\mathbf{x}_t=\mathbf{x}} \\
&= \alpha(t)\log p_t(\mathbf{x}) - \beta(t)\mathbb{E}_{\mathbf{x}_T \sim q_{T|t}(\mathbf{x}_T|\mathbf{x})}\left[r(\mathbf{x}_T)|\mathbf{x}_t = \mathbf{x}\right] \\
&\qquad\qquad\qquad\qquad + D_{KL}\left(q(\mathbf{x}_T|\mathbf{x}_t)\|p(\mathbf{x}_T|\mathbf{x}_t)\right)\Big|_{\mathbf{x}_t=\mathbf{x}}
\end{aligned}
$$

*Proof.* We begin by rewriting the value function in terms of the diffusion posterior $p(\mathbf{x}_T \mid \mathbf{x}_t)$, which denotes the terminal distribution evolved from $\mathbf{x}_t$ under the uncontrolled process $\mathbb{P}$.

$$
\begin{aligned}
V\left(\mathbf{x}, t\right) &= -\log \frac{p_t^{\mathbf{u}}(\mathbf{x})}{p_t(\mathbf{x})} \\
&= -\log \mathbb{E}_{\mathbb{P}}\left[p_t^{-\alpha(t)}(\mathbf{x})\exp\left(\beta(t)r(\mathbf{x}_T)\right)|\mathbf{x}_t = \mathbf{x}\right] \\
&= -\log \mathbb{E}_{p(\mathbf{x}_T|\mathbf{x})}\left[p_t^{-\alpha(t)}(\mathbf{x})\exp\left(\beta(t)r(\mathbf{x}_T)\right)|\mathbf{x}_t = \mathbf{x}\right] \\
&= -\log \int p_t^{-\alpha(t)}(\mathbf{x})\exp\left(\beta(t)r(\mathbf{x}_T)\right)p(\mathbf{x}_T|\mathbf{x})d\mathbf{x}_T, \quad \textit{note: } p_t^{\mathbf{u}}(\mathbf{x}) \textit{ depends implicitly on } \mathbf{x}_T \\
&= -\log \int p_t^{-\alpha(t)}(\mathbf{x})\exp\left(\beta(t)r(\mathbf{x}_T)\right)\frac{p(\mathbf{x}_T|\mathbf{x})}{q(\mathbf{x}_T|\mathbf{x})}q(\mathbf{x}_T|\mathbf{x})d\mathbf{x}_T, \quad \textit{where } q(\mathbf{x}_T|\mathbf{x}) \textit{ is a traceable simpler distribution.} \\
&= -\log \mathbb{E}_{q(\mathbf{x}_T|\mathbf{x})}\left[p_t^{-\alpha(t)}(\mathbf{x})\exp\left(\beta(t)r(\mathbf{x}_T)\right)\frac{p(\mathbf{x}_T|\mathbf{x})}{q(\mathbf{x}_T|\mathbf{x})}|\mathbf{x}_t = \mathbf{x}\right] \\
&\leq -\mathbb{E}_{\mathbf{x}_T \sim q_{T|t}(\mathbf{x}_T|\mathbf{x})}\left[\log\left(p_t^{-\alpha(t)}(\mathbf{x})\exp\left(\beta(t)r(\mathbf{x}_T)\right)\right)|\mathbf{x}_t = \mathbf{x}\right] \\
&\qquad + D_{KL}\left(q(\mathbf{x}_T|\mathbf{x}_t)\|p(\mathbf{x}_T|\mathbf{x}_t)\right)\Big|_{\mathbf{x}_t=\mathbf{x}}, \textit{Jensen's inequality} \\
&= \alpha(t)\log p_t(\mathbf{x}) - \beta(t)\mathbb{E}_{\mathbf{x}_T \sim q_{T|t}(\mathbf{x}_T|\mathbf{x})}\left[r(\mathbf{x}_T)|\mathbf{x}_t = \mathbf{x}\right] + D_{KL}\left(q(\mathbf{x}_T|\mathbf{x}_t)\|p(\mathbf{x}_T|\mathbf{x}_t)\right)\Big|_{\mathbf{x}_t=\mathbf{x}} \\
&= \bar{V}(\mathbf{x}, t, q)
\end{aligned}
\tag{12}
$$

**Lemma 3.3** *For $\mathbf{x}_T \sim p(\mathbf{x}_T|\mathbf{x}_t)$, the data posterior score $\nabla_{\mathbf{x}_T} \log p(\mathbf{x}_T|\mathbf{x}_t)$ can be approximated in terms of the model scores $\mathbf{s}_\theta(\cdot)$ as follows:*

$$\nabla_{\mathbf{x}_T} \log p(\mathbf{x}_T|\mathbf{x}_t) \approx \mathbf{s}_\theta(\mathbf{x}_T) - \eta(t)\mathbf{s}_\theta(\mathbf{x}_t)$$

*Proof.* We commence by expressing the marginal data score as the expectation of the conditional data score over the posterior distribution.

$$
\begin{aligned}
\nabla_{\mathbf{x}_t} \log p(\mathbf{x}_t) &= \nabla_{\mathbf{x}_t} \log \int p(\mathbf{x}_t, \mathbf{x}_T) d\mathbf{x}_T \\
&= \nabla_{\mathbf{x}_t} \log \int p(\mathbf{x}_t \mid \mathbf{x}_T) p(\mathbf{x}_T) d\mathbf{x}_T \\
&= \frac{1}{p(\mathbf{x}_t)} \nabla_{\mathbf{x}_t} \int p(\mathbf{x}_t \mid \mathbf{x}_T) p(\mathbf{x}_T) d\mathbf{x}_T, \ \textit{Identity's rule} \\
&= \frac{1}{p(\mathbf{x}_t)} \int \nabla_{\mathbf{x}_t} p(\mathbf{x}_t \mid \mathbf{x}_T) p(\mathbf{x}_T) d\mathbf{x}_T \\
&= \frac{1}{p(\mathbf{x}_t)} \int p(\mathbf{x}_T, \mathbf{x}_t) \nabla_{\mathbf{x}_t} \log p(\mathbf{x}_t \mid \mathbf{x}_T) d\mathbf{x}_T \\
&= \frac{1}{p(\mathbf{x}_t)} \int p(\mathbf{x}_t \mid \mathbf{x}_T) \nabla_{\mathbf{x}_t} \log p(\mathbf{x}_t \mid \mathbf{x}_T) p(\mathbf{x}_T) d\mathbf{x}_T, \ \textit{Identity's rule} \\
&= \int p(\mathbf{x}_T \mid \mathbf{x}_t) \nabla_{\mathbf{x}_t} \log p(\mathbf{x}_t \mid \mathbf{x}_T) d\mathbf{x}_T \\
&= E_{p(\mathbf{x}_T|\mathbf{x}_t)}[\nabla_{\mathbf{x}_t} \log p(\mathbf{x}_t|\mathbf{x}_T)]
\end{aligned}
\tag{13}
$$

We leverage the identity, $\nabla_x \log f(x) = \frac{1}{f(x)}\nabla_x f(x)$, in two critical steps. Applying a one-sample Monte Carlo estimation of the expectation, we obtain the following approximation.

$$s_\theta(\mathbf{x}_t) = \nabla_{\mathbf{x}_t} \log p(\mathbf{x}_t) \approx \nabla_{\mathbf{x}_t} \log p(\mathbf{x}_t|\mathbf{x}_T), \qquad \mathbf{x}_T \sim p(\mathbf{x}_T|\mathbf{x}_t) \tag{14}$$

Assuming the noise kernel with the form $p_{t|T}(\mathbf{x}_t|\mathbf{x}_T) = \mathcal{N}(\eta(t)\mathbf{x}_T, \gamma^2(t)I)$, the model score can be expressed as:

$$s_\theta(\mathbf{x}_t) \approx \nabla_{\mathbf{x}_t} \log p(\mathbf{x}_t|\mathbf{x}_T) = -\frac{\mathbf{x}_t - \eta(t)\mathbf{x}_T}{\gamma^2(t)} = -\frac{\mathbf{z}}{\gamma(t)}, \qquad \mathbf{z} \sim \mathcal{N}(0, I) \tag{15}$$

The last equality results from $\mathbf{x}_t = \eta(t)\mathbf{x}_T + \gamma(t)\mathbf{z}, \quad \mathbf{z} \sim \mathcal{N}(0, I)$. Moreover, the posterior score can be decomposed as:

$$
\begin{aligned}
\nabla_{\mathbf{x}_T} \log p(\mathbf{x}_T|\mathbf{x}_t) &= \nabla_{\mathbf{x}_T} \log \frac{p(\mathbf{x}_t|\mathbf{x}_T)p(\mathbf{x}_T)}{p(\mathbf{x}_t)} \\
&= \nabla_{\mathbf{x}_T} \log p(\mathbf{x}_t|\mathbf{x}_T) + \nabla_{\mathbf{x}_T} \log p(\mathbf{x}_T) \\
&= \eta(t)\frac{\mathbf{x}_t - \eta(t)\mathbf{x}_T}{\gamma^2(t)} + s_\theta(\mathbf{x}_T) \\
&= \eta(t)\frac{\mathbf{z}}{\gamma(t)} + s_\theta(\mathbf{x}_T) \\
&\approx s_\theta(\mathbf{x}_T) - \eta(t)s_\theta(\mathbf{x}_t)
\end{aligned}
\tag{16}
$$

We approximate the last equation with the model score from Equation 15, and sample $\mathbf{x}_T \sim p(\mathbf{x}_T|\mathbf{x}_t)$ as in Equation 13, then we conclude the proof.

**Proposition 3.4** *Consider the low-density reward-based cost functional $\widetilde{J}(\mathbf{u}, \mathbf{x}, t)$ and its upper bound value function $\bar{V}(\mathbf{x}, t, q)$. Let $q_{T|t}(\mathbf{x}_T|\mathbf{x}_t)$ denote the proposal posterior initialized via Tweedie's formula, and let $q^{\epsilon}_{T|t}(\mathbf{x}_T|\mathbf{x}_t)$ denote the updated posterior obtained after applying the back-and-forth Stein correction with step size $\epsilon(t)$. Then, the optimal control $\bar{\mathbf{u}}^*(\mathbf{x}, t)$ for the surrogate value function $\bar{V}(\mathbf{x}, t, q)$ decomposed as:*

$$
\frac{\bar{\mathbf{u}}^*(\mathbf{x}_t^i, t)}{\sigma(t)} = \underbrace{-\alpha(t)\mathbf{s}_\theta(\mathbf{x}_t^i) + \beta(t)\nabla_{\mathbf{x}_t^i}\mathbb{E}_{\mathbf{x}_T^i \sim q^{\epsilon}_{T|t}(\mathbf{x}_T|\mathbf{x}_t)}\Big[r(\mathbf{x}_T^i)\Big]}_{\text{Low-density reward-based guidance on } \mathcal{M}_t}
$$

$$
\oplus \quad \underbrace{\mathbb{E}_{\mathbf{x}_T^j \sim q_{T|t}(\mathbf{x}_T|\mathbf{x}_t)}\Big[\Big(\mathbf{s}_\theta(\mathbf{x}_T^j) - \eta(t)\mathbf{s}_\theta(\mathbf{x}_t^j)\Big)k(\mathbf{x}_T^i, \mathbf{x}_T^j) + \nabla_{\mathbf{x}_T^j}k(\mathbf{x}_T^i, \mathbf{x}_T^j)\Big]}_{\text{Training-free Stein diffusion posterior correction on } \mathcal{M}_T}
$$

*Proof.* By Lemma 2.2, we obtain the optimal control of the surrogate value function.

$$
\frac{\bar{\mathbf{u}}^*(\mathbf{x}_t^i, t)}{\sigma(t)} = -\nabla_{\mathbf{x}_t^i}\bar{V}(\mathbf{x}_t^i, t, q)
$$
$$
= \underbrace{-\nabla_{\mathbf{x}_t^i}D_{KL}\Big(q(\mathbf{x}_T^i|\mathbf{x}_t^i)\|p(\mathbf{x}_T^i|\mathbf{x}_t^i)\Big)}_{\mathbf{I}} + \underbrace{-\alpha(t)\mathbf{s}_\theta\Big(\mathbf{x}_t^i\Big) + \beta(t)\nabla_{\mathbf{x}_t^i}\mathbb{E}_{\mathbf{x}_T^i \sim q_{T|t}(\mathbf{x}_T|\mathbf{x}_t)}\Big[r(\mathbf{x}_T^i)\Big]}_{\mathbf{II}} \tag{17}
$$

The first control component (I) guides posterior samples in the direction that minimizes the KL divergence between the approximate and true posteriors. The second control component (II) enables low-density sampling and reward/classifier-based diffusion guidance. Most existing training-free diffusion guidance methods utilize solely the second control component while ignoring the first one, which thus does not guarantee sampling from the true posterior. In this work, we propose Stein Diffusion Guidance (SDG), which incorporates the guidance control from both components. Let's consider a proposal posterior $q_{T|t}(\mathbf{x}_T|\mathbf{x}_t)$, whose mean value is estimated via Tweedie's formula $(\mathbf{x}_t + \gamma^2(t)\mathbf{s}_\theta(\mathbf{x}_t, t))/\eta(t)$. Below, we present the analytical form of each control component.

*The KL divergence control* (I): Since the true posterior $p(\mathbf{x}_T^i|\mathbf{x}_t^i)$ does not admit a closed-form expression, we can not compute analytically the KL divergence and its gradient. To address this, we leverage the Stein variational inference, a nonparametric approach that identifies the steepest gradient direction to minimize the KL divergence. Assuming a batch of N particles at the $t^{th}$ reverse diffusion timestep $\mathcal{D}_t \leftarrow \{\mathbf{x}_t^i\}_{i=0}^N$, we apply Lemma 2.1 to obtain the KSD's direction minimizing $D_{KL}\Big(q(\mathbf{x}_T^i|\mathbf{x}_t^i)\|p(\mathbf{x}_T^i|\mathbf{x}_t^i)\Big)$:

$$
\phi^*(\mathbf{x}_t^i) = \mathbb{E}_{\mathbf{x}_T^j \sim q_{T|t}(\mathbf{x}_T|\mathbf{x}_t)}[\nabla_{\mathbf{x}_T^j}\log p(\mathbf{x}_T^j|\mathbf{x}_t^j)k(\mathbf{x}_T^i, \mathbf{x}_T^j) + \nabla_{\mathbf{x}_T^j}k(\mathbf{x}_T^i, \mathbf{x}_T^j)]
$$

Computing this optimal direction requires numerous Jacobian-vector product evaluations, e.g., $\nabla_{\mathbf{x}_T^j}\log p(\mathbf{x}_T^j|\mathbf{x}_t^j)\frac{\partial \mathbf{x}_T^j}{\partial \mathbf{x}_t^j}$ and $\nabla_{\mathbf{x}_T^j}k(\mathbf{x}_T^i, \mathbf{x}_T^j)\frac{\partial \mathbf{x}_T^j}{\partial \mathbf{x}_t^j}$, which are computationally expensive as the number of particles $N$ increases. To alleviate this computational burden, we propose *a back-and-forth Stein correction*: (**i**) Apply Tweedie's formula to map backward the particles $\mathcal{D}_t \leftarrow \{\mathbf{x}_t^i\}_{i=0}^N$ on $\mathcal{M}_t$ to $\mathcal{D}_T \leftarrow \Big\{\mathbf{x}_T^i|\mathbf{x}_T^i = \frac{\mathbf{x}_t^i+\gamma^2(t)\mathbf{s}_\theta(\mathbf{x}_t^i)}{\eta(t)}, \forall \mathbf{x}_t^i \in \mathcal{D}_t\Big\}$ on $\mathcal{M}_T$, which represents the initial proposal posterior $q_{T|t}(\mathbf{x}_T|\mathbf{x}_t)$; (**ii**) Apply the Stein correction on the particles of $\mathcal{D}_T$, which results in the corrected posterior $q^{\epsilon}_{T|t}(\mathbf{x}_T|\mathbf{x}_t)$; (**iii**) Apply the perturbation kernel $p_{t|T}(\mathbf{x}_t|\mathbf{x}_T)$ to map forward the Stein-corrected particles $\mathcal{D}_T$ on $\mathcal{M}_T$ to $\mathcal{D}_t \leftarrow \Big\{\mathbf{x}_i^t|\mathbf{x}_i^t \sim \mathcal{N}(\eta(t)\mathbf{x}_T^i, \gamma^2(t)I), \forall \mathbf{x}_T^i \in \mathcal{D}_T\Big\}$ on $\mathcal{M}_t$. In the second step, the Stein correction applies a particle-based transform on each particle of $\mathcal{D}_T$, which follows the KSD's direction, given as:

$$
\phi^*(\mathbf{x}_T^i) = \mathbb{E}_{\mathbf{x}_T^j \sim q_{T|t}(\mathbf{x}_T|\mathbf{x}_t)}[\nabla_{\mathbf{x}_T^j}\log p(\mathbf{x}_T^j|\mathbf{x}_t^j)k(\mathbf{x}_T^i, \mathbf{x}_T^j) + \nabla_{\mathbf{x}_T^j}k(\mathbf{x}_T^i, \mathbf{x}_T^j)] \tag{18}
$$

We observe that the Stein correction on $\mathcal{M}_T$ does not involve any evaluations of Jacobian-vector products, which achieves more memory efficiency during inference. By replacing the posterior score from Lemma 3.3, we obtain the KSD's direction with a closed form.

$$\phi^*(\mathbf{x}_T^i) = \mathbb{E}_{\mathbf{x}_T^j \sim q_{T|t}(\mathbf{x}_T|\mathcal{D}_t)}[(-\eta(t)\mathbf{s}_\theta(\mathbf{x}_t^j) + \mathbf{s}_\theta(\mathbf{x}_T^j))k(\mathbf{x}_T^i, \mathbf{x}_T^j) + \nabla_{\mathbf{x}_T^j} k(\mathbf{x}_T^i, \mathbf{x}_T^j)] \tag{19}$$

The particle update with a stepsize $\epsilon(t)$ can be taken as:

$$\mathbf{x}_T^i = \mathbf{x}_T^i + \epsilon(t) * \phi^*(\mathbf{x}_T^i) \tag{20}$$

*The low-density reward guidance control* (II): We use the Stein-corrected posterior $q_{T|t}^\epsilon(\mathbf{x}_T|\mathbf{x}_t)$ to guide samples toward low-density regions with high rewards or desired properties. The optimal control for this task can be written in an analytical form.

$$-\alpha(t)\mathbf{s}_\theta\left(\mathbf{x}_t^i\right) + \beta(t)\nabla_{\mathbf{x}_t^i}\mathbb{E}_{\mathbf{x}_T^i \sim q_{T|t}^\epsilon(\mathbf{x}_T|\mathbf{x}_t)}\left[r(\mathbf{x}_T^i)\right] \tag{21}$$

Where $\mathbf{x}_t^i$ is the noised sample on $\mathcal{M}_t$, resulting from applying the forward kernel on its Stein-corrected version $\mathbf{x}_T^i$ on $\mathcal{M}_T$.

By concatenating the two control components, we conclude the proof.

**Corollary 3.5**    *Let the correction stepsize be set to zero, i.e., $\epsilon(t) = 0$ for all reverse steps $t$. Then, the back-and-forth Stein correction generalizes the Langevin correction as a special case with stepsize $\gamma^2(t)$ and noise scaled by $\sqrt{2}$:*

$$\mathbf{x}_t \leftarrow \mathbf{x}_t + \gamma^2(t)s_\theta(\mathbf{x}_t) + \gamma(t)\mathbf{z}, \qquad \mathbf{z} \sim \mathcal{N}(0, I)$$

*Proof.* By setting $\epsilon(t) = 0$ for all $t$, from Algorithm 1, we have the back-and-forth Stein correction mechanism with a following analytical form:

$$\begin{aligned}\mathbf{x}_t &\leftarrow \eta(t)\frac{\mathbf{x}_t + \gamma^2(t)s_\theta(\mathbf{x}_t)}{\eta(t)} + \gamma(t)\mathbf{z} \\ &= \mathbf{x}_t + \gamma^2(t)s_\theta(\mathbf{x}_t) + \gamma(t)\mathbf{z}, \qquad \mathbf{z} \sim \mathcal{N}(0, I)\end{aligned} \tag{22}$$

This corresponds to the Langevin correction from Song et al. (2020b), with a step size of $\gamma^2(t)$ and the noise term scaled down by a factor of $\sqrt{2}$. As a result, the back-and-forth Stein correction generalizes the Langevin correction as a special case.

## C. Molecular Sampling in Low-Density Regions

### C.1. Sampling Molecular Graph Permutation-Invariant Distributions

Our target application of Stein Diffusion Guidance is to enable sampling molecular graphs with desired rewards in low-density regions in a training-free diffusion guidance manner. We adapt the score-based generative framework from Jo et al. (2022) for modeling molecular graph distributions. Given a graph representation $\mathcal{G} = (\mathbf{X}, \mathbf{E})$, with $\mathbf{X}$ and $\mathbf{E}$ denoting the node and edge feature matrices, respectively, the authors introduce a *system* of SDEs to capture molecular graph distributions, whose reverse/generative process can be derived as:

$$\begin{cases} \mathbf{X}_{t+1} = \mathbf{X}_t - \mathbf{b}_{\mathbf{X}}(\mathbf{X}_t, t) + \sigma_{\mathbf{X}}(t)^2 s_\theta(\mathbf{X}_t, \mathbf{E}_t) + \sigma_{\mathbf{X}}(t)\mathbf{Z}, & \mathbf{Z} \sim \mathcal{N}(0, I) \\ \mathbf{E}_{t+1} = \mathbf{E}_t - \mathbf{b}_{\mathbf{E}}(\mathbf{E}_t, t) + \sigma_{\mathbf{E}}(t)^2 s_\phi(\mathbf{X}_t, \mathbf{E}_t) + \sigma_{\mathbf{E}}(t)\mathbf{Z}, & \mathbf{Z} \sim \mathcal{N}(0, I) \end{cases} \tag{23}$$

The authors utilize two score networks to approximate conditional data scores: $s_\theta(\mathbf{X}_t, \mathbf{E}_t) \approx \nabla_{\mathbf{X}_t} \log p(\mathbf{X}_t, \mathbf{E}_t)$ and $s_\phi(\mathbf{X}_t, \mathbf{E}_t) \approx \nabla_{\mathbf{E}_t} \log p(\mathbf{X}_t, \mathbf{E}_t)$. In addition, $s_{\theta,\phi}(\mathbf{X}_t, \mathbf{E}_t)$ are permutation-equivariant models that respect the inherent symmetry of graph data. This system of coupled SDEs must be solved simultaneously in order to sample graph distributions. Building on this unconditional sampling foundation, we extend Stein Diffusion Guidance to sample molecular graphs with desired properties in low-density regions. From Algorithm 1, SDG first corrects posterior molecular graph samples on the manifold $\mathcal{M}_T$:

$$\begin{cases} \mathbf{X}_T^i = \mathbf{X}_T^i + \epsilon_{\mathbf{X}}(t) \frac{1}{N} \sum_{\mathbf{X}_t^j, \mathbf{X}_T^j \in \mathcal{D}_t \bigcup \mathcal{D}_T} \Big( (s_\theta(\mathbf{X}_T^j, \mathbf{E}_T^j) - \eta_{\mathbf{X}}(t) s_\theta(\mathbf{X}_t^j, \mathbf{E}_t^j)) k(\mathbf{X}_T^i, \mathbf{X}_T^j) + \nabla_{\mathbf{X}_T^j} k(\mathbf{X}_T^i, \mathbf{X}_T^j) \Big) \\ \mathbf{E}_T^i = \mathbf{E}_T^i + \epsilon_{\mathbf{E}}(t) \frac{1}{N} \sum_{\mathbf{E}_t^j, \mathbf{E}_T^j \in \mathcal{D}_t \bigcup \mathcal{D}_T} \Big( (s_\phi(\mathbf{X}_T^j, \mathbf{E}_T^j) - \eta_{\mathbf{E}}(t) s_\phi(\mathbf{X}_t^j, \mathbf{E}_t^j)) k(\mathbf{E}_T^i, \mathbf{E}_T^j) + \nabla_{\mathbf{E}_T^j} k(\mathbf{E}_T^i, \mathbf{E}_T^j) \Big) \end{cases} \quad (24)$$

And then utilizing the Stein-corrected posterior samples to perform low-density diffusion guidance with off-the-shelf molecular property predictors; we refer to property predictors as classifiers.

$$\begin{cases} \mathbf{X}_{t+1}^i = \mathbf{X}_t^i - \mathbf{b}_{\mathbf{X}}(\mathbf{X}_t^i, t) + \sigma_{\mathbf{X}}(t)^2 \Big( (1 - \alpha_{\mathbf{X}}(t)) s_\theta(\mathbf{X}_t^i, \mathbf{E}_t^i) + \beta_{\mathbf{X}}(t) \nabla_{\mathbf{X}_t^i} r(\mathbf{X}_T^i, \mathbf{E}_T^i) \Big) + \sigma_{\mathbf{X}}(t)\mathbf{Z} \\ \mathbf{E}_{t+1}^i = \mathbf{E}_t^i - \mathbf{b}_{\mathbf{E}}(\mathbf{E}_t^i, t) + \sigma_{\mathbf{E}}(t)^2 \Big( (1 - \alpha_{\mathbf{E}}(t)) s_\phi(\mathbf{X}_t^i, \mathbf{E}_t^i) + \beta_{\mathbf{E}}(t) \nabla_{\mathbf{E}_t^i} r(\mathbf{X}_T^i, \mathbf{E}_T^i) \Big) + \sigma_{\mathbf{E}}(t)\mathbf{Z} \end{cases} \quad (25)$$

We obtain initial posterior samples via Tweedie's formula, given as $\mathbf{X}_T^i = (\mathbf{X}_t^i + \gamma_{\mathbf{X}}(t) s_\theta(\mathbf{X}_t^i, \mathbf{E}_t^i)) / \eta_{\mathbf{X}}(t)$ and $\mathbf{E}_T^i = (\mathbf{E}_t^i + \gamma_{\mathbf{E}}(t) s_\phi(\mathbf{X}_t^i, \mathbf{E}_t^i)) / \eta_{\mathbf{E}}(t)$.

Based on the primary work (Jo et al., 2022), Lee et al. (2023) further propose a standard classifier-based diffusion guidance for conditional molecular sampling in out-of-distribution settings. In experiments, we use the same pretrained models and settings as Lee et al. (2023). Concretely, we model the node component using a Variance Preserving SDE (VPSDE) and the edge component using a Variance Exploding SDE (VESDE). In sampling, we adopt the predictor-corrector scheme, with the reverse SDE as the predictor and annealed Langevin dynamics as the corrector. As reported in Jo et al. (2022) (Table 12), the pretrained models tend to sample molecules with very low validity when using either the predictor or corrector framework alone.

### C.2. Experimental Setup

**Datasets.** We benchmark SDG on molecular generation tasks to discover novel ligands with strong binding affinity for specific protein targets. Following Lee et al. (2023), we evaluate performance on three protein receptors: **Fa7** (Coagulation factor VII), **5ht1b** (5-hydroxytryptamine receptor 1B), **Jak2** (Tyrosine-protein kinase JAK2), and **Parp1** (Poly[ADP-ribose] polymerase-1). Ligand candidates are sampled from the learned distribution over the ZINC250k dataset (Irwin et al., 2012). To assess binding affinity, we compute docking scores using the program QuickVina 2 (Alhossary et al., 2015) and set the exhaustiveness to 1 by following Lee et al. (2023).

**Evaluation Metrics.** We assess ligand-protein docking based on four criteria: (1) Tanimoto similarity (**SIM**) with the closest ZINC250k training sample is below 0.4 to ensure novelty (**Nov.**); (2) synthetic accessibility score (**SA**) is below 5, indicating ease of synthesis; (3) drug-likeness score (**QED**) exceeds 0.5; and (4) docking score (**DS**) is lower than the median DS of known actives. For consistent comparison and training, each raw score is normalized to lie within the range, $0 \le \hat{\mathbf{SA}}, \hat{\mathbf{DS}} \le 1$:

$$\hat{\mathbf{SA}} = \frac{10 - \mathbf{SA}}{9} \qquad\qquad \hat{\mathbf{DS}} = \frac{\mathbf{DS}}{\min(\mathbf{DS}_{train}) - 0.2} \qquad (26)$$

Where higher values indicate better performance. The overall evaluation metric, **Novel Hit Ratio (%)**, is the percentage of unique (**Uniq.**) molecules among 3,000 samples that satisfy these criteria.

**Model and Baselines.** We adopt the pretrained score-based generative model GDSS (Jo et al., 2022) and its classifier-guided variant MOOD (Lee et al., 2023) as baselines. We also compare SDG with several non-diffusion baselines: FREED (Yang et al., 2021), a fragment-based reinforcement learning method; HierVAE (Jin et al., 2020), a VAE model with a hierarchical molecular representation; and MORLD (Jeon & Kim, 2020), a reinforcement learning approach that incorporates QED, SA, and DS at different optimisation stages. We also report ablations of SDG: *SDG♣ w/o Stein correction*,

corresponding to standard training-free diffusion guidance; $SDG^\heartsuit$ ($\alpha(t) = 0$, $\epsilon(t) > 0$), corresponding to guidance without low-density sampling; and $SDG^\diamondsuit$ ($\alpha(t) > 0$, $\epsilon(t) = 0$), corresponding to the Langevin correction (Cororally 3.5).

**Novel Diffusion Prior on the Order of Molecule Graphs**   In graph generative modeling, the sampling process often utilizes the marginal graph order data distribution. However, for a certain type of molecular properties, the distribution of desired molecules over graph order is usually nonuniform; i.e., molecules can behave differently in molecular property space according to their node cardinality. Here, we propose a novel prior on the order of graphs that prioritizes sampling graph order, whose training molecules exhibit desired docking scores on a target protein receptor.

$$p^\dagger(N_i) = \frac{\left| N_i \cap (\mathbf{DS}_i < \tau) \right| + M \times 39}{M \times 39 + \sum_i N_i} \times \frac{p(N_i)}{p(\mathbf{DS} < \tau)} \tag{27}$$

Where $\left| \cdot \right|$ denotes the cardinality of set satisfying the $N_i$ number of nodes and their docking scores $\mathbf{DS}_i$ below the hit threshold $\tau$, i.e, $\left| N_i \cap (\mathbf{DS}_i < \tau) \right|$; M denotes the offset number that serves to enable sampling the graph order which does not have any molecules within desired docking scores; we choose $M = 10$ for all experiments; in ZINC250k dataset, the range of graph order is from 0 to 38, which results to 39 different possibilities; $p(N_i)$ is the marginal graph order distribution computed from training data; and $p(\mathbf{DS} < \tau)$ denotes the marginal distribution of hit training molecules.

**Property Predictor Pretraining on Clean Data.**   Since there are no available predictors for multiple target properties, we opt to pre-train our molecular property predictors on clean (i.e., noise-free) molecular data of ZINC250k, where the target property is defined as the product of the normalized scores: $\hat{\mathbf{SA}} \times \hat{\mathbf{DS}}$. Since most training molecules satisfy the **QED** hit condition, we thus ignore this target to simplify our multi-objective optimization task. Our regressor architecture is similar to the one from Lee et al. (2023) with an additional graph convolution layer. We set the learning rate to 0.01, the number of epochs to 10, the AdamW optimizer (Loshchilov & Hutter, 2019), and utilize the same architecture hyperparameters for all target proteins.

**Stein Correction Stepsize.**   We utilize an adaptive stepsize schedule $\epsilon(t)$ similar to the corrector framework from Song et al. (2020b), which is defined as:

$$\epsilon(t) = 2\eta^2(t)\Big(snr\|\mathbf{z}\|_2/\|\mathbf{g}\|_2\Big)^2, \qquad \mathbf{z} \sim \mathcal{N}(0, I) \tag{28}$$

where $snr$ denotes the signal-to-noise ratio, and $\mathbf{g} = \mathbf{s}_\theta\Big(\mathbf{x}_T^j\Big) - \eta(t)\mathbf{s}_\theta\Big(\mathbf{x}_t^j\Big)$, assuming a forward noising kernel as $p_{t|T}(\mathbf{x}_t|\mathbf{x}_T) = \mathcal{N}(\eta(t)\mathbf{x}_T, \gamma^2(t)I)$.

**Annealing Schedules.**   For low-density sampling, we experiment on two scheduling approaches: *constant* and *linear*.

$$\alpha(t) = \left\{ \begin{array}{ll} \alpha_{max}, & constant \\ t \times \alpha_{max}, & linear \end{array} \right. \tag{29}$$

For guidance strength, we adopt the Polyak stepsize (Hazan & Kakade, 2019; Shen et al., 2024):

$$\beta(t) = \beta_{max} \times \frac{\|s_\theta(\mathbf{x}_t)\|}{\|\nabla_{\mathbf{x}_t} r(\mathbf{x}_T)\|} \tag{30}$$

**Hyperparameter Search.**   We conducted a hyperparameter search on SDG. We set the signal noise ratio $snr = \{0.2, 0.3, 0.35\}$, the maximum low-density level $\alpha = \{0.1, 0.2, 0.35, 0.42\}$, the maximum guidance strength $\beta = \{0.5, 0.7, 1.0\}$, the number of particles $N = \{512, 1024, 3000\}$, and the anneal scheduling $\alpha_{\text{scheduling}} = \{\text{linear}, \text{constant}\}$. Table 5 presents the hyperparameters of the main experimental results.

*Table 5.* SDG training hyperparameters.

|                        | Fa7      | Jak2     | 5ht1b    | Parp1    |
|------------------------|----------|----------|----------|----------|
| $snr_{\mathbf{X}}$     | 0.2      | 0.35     | 0.3      | 0.2      |
| $snr_{\mathbf{A}}$     | 0.2      | 0.35     | 0.3      | 0.2      |
| $\alpha$ scheduling    | *constant* | *constant* | *constant* | *constant* |
| $\alpha_{\mathbf{X}_{max}}$ | 0.20 | 0.42     | 0.35     | 0.1      |
| $\alpha_{\mathbf{A}_{max}}$ | 0.20 | 0.42     | 0.35     | 0.3      |
| $\beta_{\mathbf{X}_{max}}$  | 1.   | 1.       | .7       | 1.       |
| $\beta_{\mathbf{A}_{max}}$  | 0    | 0        | 0        | 0        |
| $N$                    | 512      | 3000     | 512      | 3000     |

*Table 6.* Mean and standard deviation of the top 5% novel docking scores (lower is better) across three sampling runs. Baseline results are taken from Lee et al. (2023).

| Method | Top 5% DS ($\downarrow$) | | | |
|--------|------|-------|------|-------|
|        | Fa7  | 5ht1b | Jak2 | Parp1 |
| HierVAE | $-6.812_{(\pm 0.274)}$ | $-8.081_{(\pm 0.252)}$ | $-8.285_{(\pm 0.370)}$ | $-9.487_{(\pm 0.278)}$ |
| MORLD   | $-6.263_{(\pm 0.165)}$ | $-7.869_{(\pm 0.650)}$ | $-7.816_{(\pm 0.133)}$ | $-7.532_{(\pm 0.260)}$ |
| FREED   | $-8.297_{(\pm 0.094)}$ | $-10.425_{(\pm .331)}$ | $-9.624_{(\pm 0.102)}$ | $-10.427_{(\pm 0.177)}$ |
| GDSS    | $-7.775_{(\pm 0.039)}$ | $-9.459_{(\pm 0.101)}$ | $-8.926_{(\pm 0.089)}$ | $-9.967_{(\pm 0.028)}$ |
| MOOD    | $-8.160_{(\pm 0.071)}$ | $-11.145_{(\pm 0.042)}$ | $-10.147_{(\pm 0.060)}$ | $-10.865_{(\pm 0.113)}$ |
| SDG♣    | $-7.794_{(\pm 0.040)}$ | $-7.370_{(\pm 0.201)}$ | $-5.904_{(\pm 0.163)}$ | $-9.583_{(\pm 0.170)}$ |
| SDG     | $-8.310_{(\pm 0.009)}$ | $-11.383_{(\pm 0.0537)}$ | $-10.178_{(\pm 0.037)}$ | $-11.088_{(\pm 0.147)}$ |

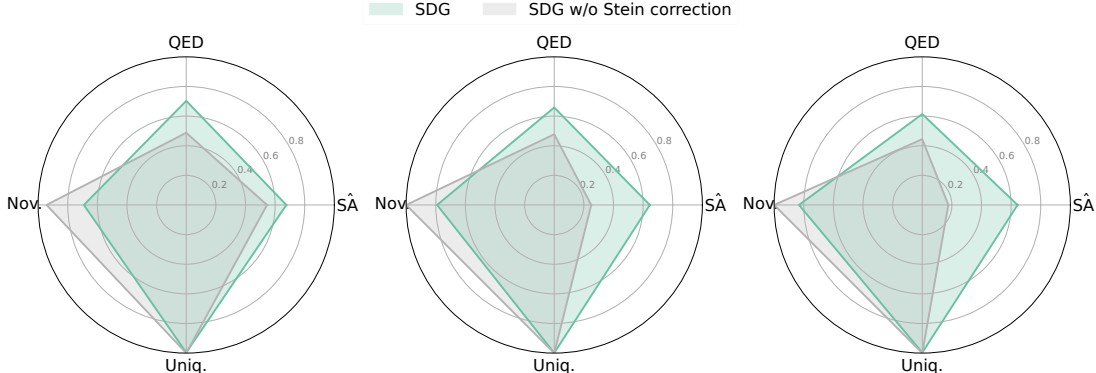

*Figure 16.* Multiple sampling objectives, including sample novelty (Nov.), drug-likeness (QED), normalized synthetic accessibility (SÂ), and sample uniqueness (Uniq.) for Fa7 (left), Jak2 (middle), 5ht1b (right). Higher scores indicate better performance.

**Hardware Usage.**   All experiments are conducted on NVIDIA Titan RTX, RTX 6000 with 8 CPU cores, using a Slurm-managed high-performance computing (HPC) system.

## C.3. Additional Results

### C.3.1. MULTI-OBJECTIVE OPTIMIZATION

Figure 16 displays the radar plots of SDG performance under multiple property constraints. A consistent pattern emerges across all target proteins: without Stein correction, SDG tends to generate novel molecules with low drug-likeness and poor synthetic, largely due to overly complex structures. Notably, SDG also achieves nearly $100\%$ molecular uniqueness.

### C.3.2. NOVEL TOP 5% DOCKING SCORES

We additionally report in Table 6 the average docking scores of the top 5% unique molecules that satisfy the novel hit conditions. As observed, SDG yields significantly lower average docking scores compared to the standard training-free

diffusion guidance method SDG♣(w/o Stein correction), indicating that the generated molecules exhibit a stronger binding affinity. Moreover, SDG improves binding affinity over both the pretrained model GDSS and the classifier-based diffusion guidance method MOOD.

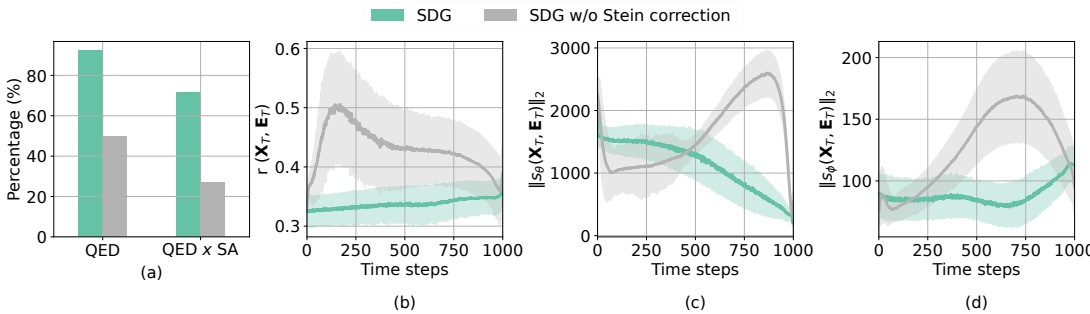

*Figure 17.* Temporal sampling dynamics of SDG for the Fa7 protein. (a) Percentage of molecules meeting QED and SA hit criteria; (b) Rewards of posterior samples $r(\mathbf{X}_T, \mathbf{E}_T)$; (c) Frobenius norm of node posterior scores $s_\theta(\mathbf{X}_T, \mathbf{E}_T)$; (d) Frobenius norm of edge posterior scores $s_\phi(\mathbf{X}_T, \mathbf{E}_T)$.

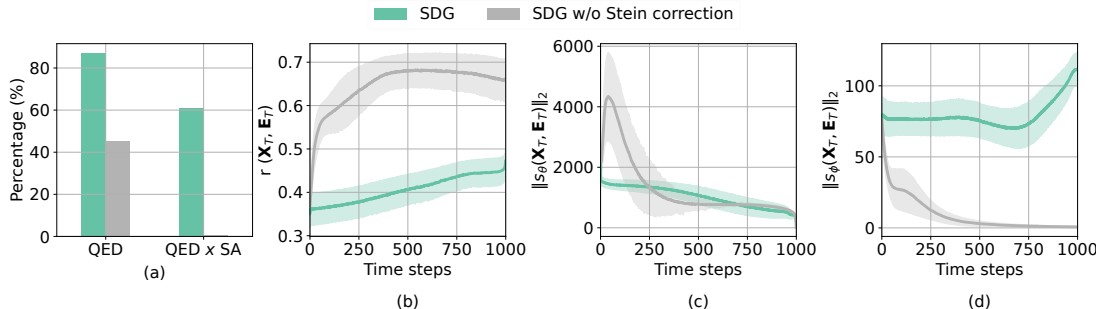

*Figure 18.* Temporal sampling dynamics of SDG for the Jak2 protein. (a) Percentage of molecules meeting QED and SA hit criteria; (b) Rewards of posterior samples $r(\mathbf{X}_T, \mathbf{E}_T)$; (c) Frobenius norm of node posterior scores $s_\theta(\mathbf{X}_T, \mathbf{E}_T)$; (d) Frobenius norm of edge posterior scores $s_\phi(\mathbf{X}_T, \mathbf{E}_T)$.

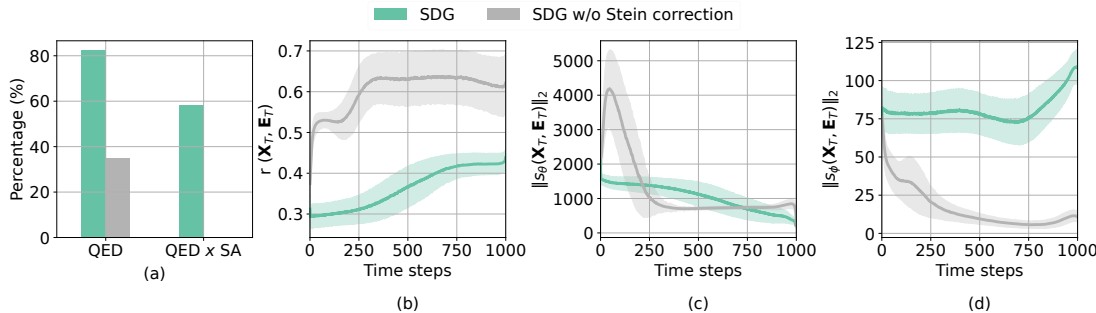

*Figure 19.* Temporal sampling dynamics of SDG for the 5ht1b protein. (a) Percentage of molecules meeting QED and SA hit criteria; (b) Rewards of posterior samples $r(\mathbf{X}_T, \mathbf{E}_T)$; (c) Frobenius norm of node posterior scores $s_\theta(\mathbf{X}_T, \mathbf{E}_T)$; (d) Frobenius norm of edge posterior scores $s_\phi(\mathbf{X}_T, \mathbf{E}_T)$.

### C.3.3. SAMPLING DYNAMICS OF MODEL SCORES ON $\mathcal{M}_T$

In the experiments, we only apply the guidance on the node's score model, $\beta_X(t) > 0$, and ignore the guidance on the edge's score model, $\beta_A(t) = 0$ for all $t$. This enables low-cost sampling by avoiding backpropagation through edge/adjacency matrices, whose dimension is $\mathcal{O}(N^2)$, with $N$ being the number of nodes. Notably, thanks to the system of coupled SDEs

(Equation 23), any updates on the node component will be appropriately reflected on the edge component as well; controlling the node-component reward guidance would thus be expressive enough to sample molecular graphs with desired properties.

During sampling process, the model scores $s(\cdot)$ tend to *increase* in norm as samples move toward the data manifold, i.e, $\|s(\mathbf{x_{t+1}})\|_2 > \|s(\mathbf{x_t})\|_2$, with $x_{t+1} \in \mathcal{M}_{t+1} \cup x_t \in \mathcal{M}_t$. In contrast, the model scores on posterior samples tend to *decrease*, i.e. $\|s(\mathbf{x}_T|\mathbf{x_{t+1}})\|_2 < \|s(\mathbf{x}_T|\mathbf{x_t})\|_2$, under the same manifold transition, reflecting that posterior samples move closer to the data distributions. These can be intuitively observed via a single data point case $\mathbf{x}^\dagger$ with Gaussian noises, where the normed conditional model score $\left\|s(\mathbf{x}, t)\right\|_2 \propto \left\|\frac{\mathbf{x}-\mathbf{x}^\dagger}{\sigma_t^2}\right\|_2$.

$$
\begin{aligned}
\text{Toward the data manifold:} \quad & t \to T, \quad & \sigma_t \to \sigma_{min}, \quad & \mathbf{x} \quad & \to \quad & \left\|s(\mathbf{x}, t)\right\|_2 \nearrow \\
\text{On the data manifold:} \quad & t = T, \quad & \sigma_{min}, \quad & \mathbf{x} \to \mathbf{x}^\dagger \quad & \to \quad & \left\|s(\mathbf{x}, t)\right\|_2 \searrow
\end{aligned}
\tag{31}
$$

Figures 17, 18, and 19(c,d) illustrate the sampling dynamics of model scores. Without the Stein correction, score dynamics fluctuate arbitrarily across both edge and node components. In contrast, with the Stein correction, node scores smoothly transit toward high-density regions, while edge scores first converge to high-density regions within the initial 700–750 steps before shifting toward lower-density regions in the opposite direction. These dynamics align well with the theoretical analysis of diffusion score behavior, providing further evidence that the Stein correction regularizes diffusion guidance in low-density regions.

### C.3.4. REWARD OVERESTIMATION IN DIFFUSION GUIDANCE

In training-free diffusion guidance, reward models and classifiers are trained only on a finite set of clean data samples, making their reward estimations reliable within the data support. However, diffusion models have much broader support due to noise injection, which can push samples outside the data support during sampling. In such regions, reward models often overestimate rewards, producing unreliable guidance. Uehara et al. (2024) were the first to formalize this issue, showing that reward models can assign excessively high scores to samples whose semantics or properties fail to meet the desired criteria. To mitigate this, they proposed regularizing the reverse diffusion process via the original stochastic optimal control formulation—though this approach is computationally expensive and impractical for efficient sampling.

Motivated by this challenge, Stein Diffusion Guidance provides a more practical alternative by solving the surrogate stochastic optimal control objective. As shown in Figures 17, 18, and 19 (a, b), the absence of Stein correction leads the reward models to overestimate genuine rewards, producing artificially inflated values. However, the corresponding molecules are often unrealistic, with significantly lower QED and SA scores. By contrast, the Stein correction introduces a low-cost regularization that mitigates reward overestimation. Furthermore, SDG-regularized model scores evolve smoothly throughout the sampling process (see Section C.3.3), keeping sampling trajectories within generative manifolds and enabling effective exploration in low-density regions.

### C.3.5. PERFORMANCE UNDER EXTREME LOW-DENSITY SAMPLING SETTINGS.

We assess the robustness of SDG under varying low-density sampling conditions by measuring the chemical (FCD) and structural (NSPDK) distances between generated and test molecules. As shown in Figures 20, 21, and 22 (a,b), removing the Stein correction causes SDG♣ to deviate substantially from the data distributions, even at modest low-density levels ($\alpha_{\max} = 0.3$). This degradation is also reflected at the molecular level, where most generated molecules exhibit invalid valency (Figures 20, 21, and 22 (c)). In contrast, with the Stein correction, SDG guides samples within generative manifolds while marginally increasing FCD and NSPDK, thereby enabling sampling from lower-density regions. Moreover, the posterior correction yields significantly more valid structures, even under extreme conditions ($\alpha_{\max} = 0.5$), underscoring the effectiveness of Stein-based regularization for robust low-density molecular sampling (Figures 20, 21, and 22 (c)).

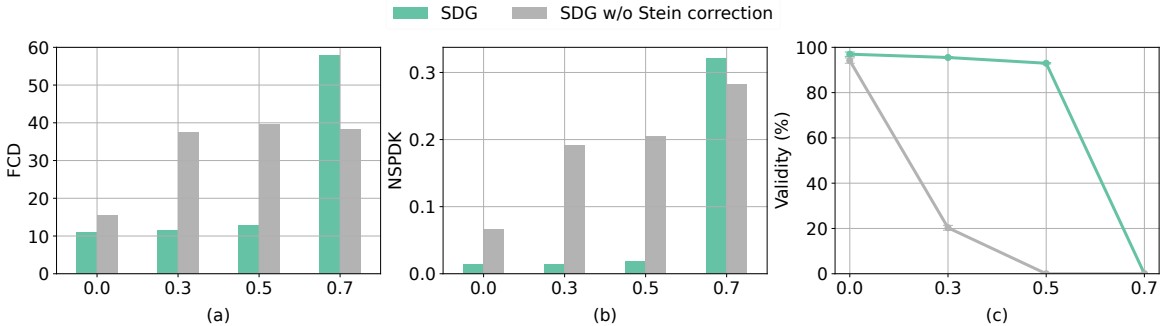

*Figure 20.* Ablation results under different low-density levels ($\alpha_{\max}$) for Fa7. Chemical distance FCD (a) and structural distance NSPDK (b) to the test set; (c) Validity of generated molecules.

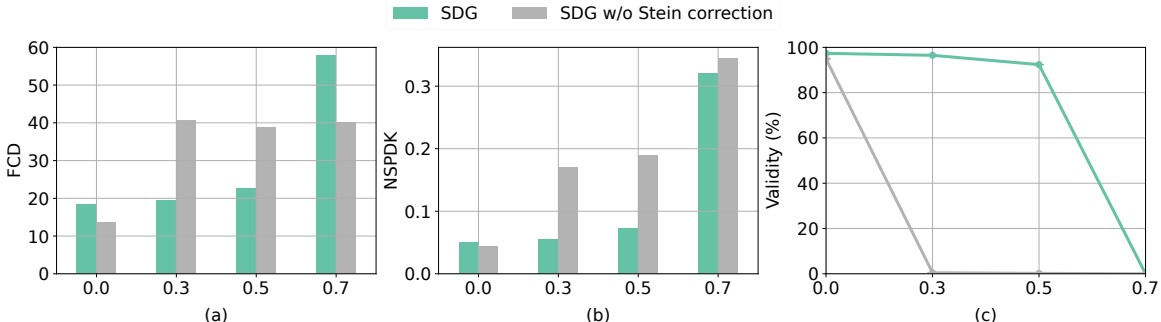

*Figure 21.* Ablation results under different low-density levels ($\alpha_{\max}$) for Jak2. Chemical distance FCD (a) and structural distance NSPDK (b) to the test set; (c) Validity of generated molecules.

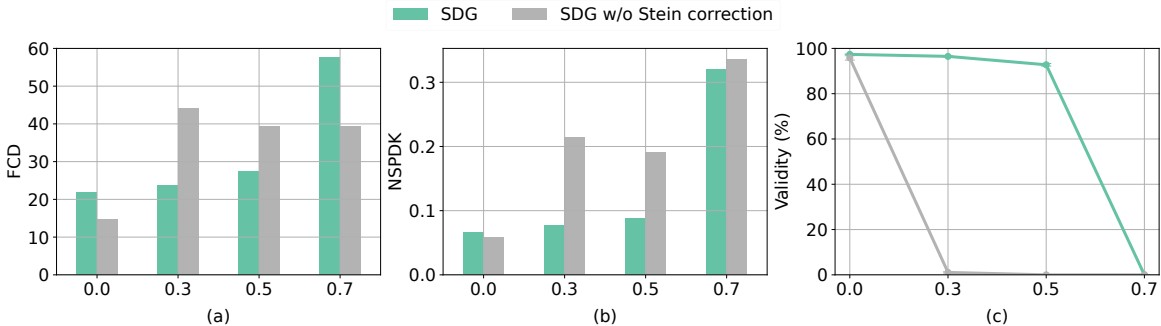

*Figure 22.* Ablation results under different low-density levels ($\alpha_{\max}$) for 5ht1b. Chemical distance FCD (a) and structural distance NSPDK (b) to the test set; (c) Validity of generated molecules.

### C.3.6. COMPUTATIONAL ANALYSIS

SDG provides a low-cost alternative to the stochastic optimal control framework of Uehara et al. (2024) for diffusion guidance sampling. Whereas the prior approach incurs a worst-case runtime complexity of $\mathcal{O}(T^2)$ due to requiring full trajectory simulations at each step, SDG achieves linear complexity $\mathcal{O}(T)$, with only a modest overhead relative to standard training-free guidance. We benchmarked on an RTX 6000 with 8 CPU cores to sample 3000 molecules, using the same solver as the pretrained score-based models (Jo et al., 2022; Lee et al., 2023). Table 7 reports SDG's computational cost in both settings. The runtime roughly doubles with the inclusion of the back-and-forth Stein correction, though this overhead can be mitigated with faster solvers such as DDIMs (Song et al., 2020a). Importantly, the back-and-forth Stein correction reduces memory overhead, yielding more efficient usage compared to the original control problem (Uehara et al., 2024).

*Table 7.* Memory efficiency analysis of the back-and-forth Stein correction on the molecule guidance task.

|  | RUNTIME (SECOND) | MEMORY (MB) |
|---|---|---|
| SDG♣ (w/o Stein correction) | 1360 | 7783 |
| SDG ($\alpha(t) > 0, \epsilon(t) > 0$) | $2521_{(\uparrow+85.4\%)}$ | $8049_{(\uparrow+3.4\%)}$ |

## D. Image Diffusion Guidance Tasks

In this section, we assert the generalizability of SDG to diffusion guidance tasks beyond molecule domains. The following experiments highlight SDG's potential applicability across different problems. Adapting the unified training-free diffusion guidance framework (TFG) of Ye et al. (2024), we evaluate SDG on four representative tasks: label guidance, Gaussian deblurring, image super-resolution, and text-to-image style transfer, whose task descriptions are presented in the table below.

*Table 8.* Task descriptions of training-free diffusion guidance on image domains.

| TASK | DESCRIPTIONS |
|---|---|
| Label guidance | Sampling images with desired class labels from CIFAR-10 (Krizhevsky et al., 2009) |
| Gaussian deblurring | Inverse problem, reconstructing the original images (Elson et al., 2007), which are blurred with Gaussian kernels. |
| Image super-resolution | Inverse problem, reconstructing the original high-resolution images (Elson et al., 2007) at $256 \times 256$ from their downsampled $64 \times 64$ counterparts. |
| Text-to-image(T2I) style transfer | Generating images that faithfully reflect input prompts while adhering to the visual style of reference images. Four style exemplars are retrieved from WikiArt (Saleh & Elgammal, 2015), and sixty-four prompts are sampled from Partiprompts (Yu et al., 2022). |

**Evaluation Metrics.**   We evaluate training-free diffusion guidance using standard image metrics. **Accuracy** (%) measures the average classification accuracy on generated samples across CIFAR-10 labels, while **FID** and **LPIPS** (Zhang et al., 2018) assess the fidelity and perceptual similarity between generated and test images. For style transfer, we evaluate performance using **Style Score** and **CLIP Score**, which measure style guidance validity and generation fidelity, respectively.

**Model Setup and Baselines.**   TFG (Ye et al., 2024) proposes a unified training-free diffusion guidance framework that integrates several existing approaches (Bansal et al., 2023; Chung et al., 2023; Song et al., 2023a; He et al., 2024) through different guidance strengths and stages. However, TFG relies on an expensive hyperparameter search, which costs approximately 2000 A100 GPU-hours, to identify an optimal combination of weights. To adapt TFG to the label guidance, Gaussian deblurring, and image super-resolution tasks, we remove the recurrent strategy, backward optimization, and manifold-preserving components (Bansal et al., 2023; He et al., 2024) (lines 5 and 8) from Algorithm 1 in Ye et al. (2024). The remaining components correspond to a standard training-free guidance approach that incorporates the implicit dynamic control mechanism (Song et al., 2023a) (line 4). Since these tasks involve no minority image classes, we set the low-density guidance factor to zero SDG♡($\alpha(t) = 0$) and focus solely on optimizing samples with highly desired properties. We also ablate with SDG♣ (w/o the Stein correction). We retain the guidance hyperparameters from TFG's parameter search. We also compare our results with DPS (Chung et al., 2023), LGD (Song et al., 2023a), UGD (Bansal et al., 2023), and MPGD (He et al., 2024). For the T2I style transfer task, we directly integrate SDG♡ into the full TFG setting by applying the back-and-forth Stein correction to diffusion posterior samples (line 6 of Algorithm 1 in TFG) before invoking the different guidance strategies (lines 7, 8, and 10 of Algorithm 1). In this case, SDG♣ corresponds to TFG's performance. Due to the high memory demands of the T2I style transfer task, we limit the SVGD update batch size to 14 particles only.

**Computation Analysis.**   We report computation details for the label guidance task. Experiments were conducted on an RTX 3090 GPU with 4 CPU cores and a batch size of 256. Table 9 summarizes the runtime and memory usage for sampling 2,560 images per class over $T = 100$ DDIM steps (Song et al., 2020a). The back-and-forth Stein correction notably reduces memory overhead. Moreover, the runtime overhead decreases compared to the previous solvers in Table 7, from $185\% = \frac{2521}{1360} \times 100$ to $136\% = \frac{820}{602} \times 100$.

*Table 9.* Computation efficiency analysis of the back-and-forth Stein correction with DDIM samplers (Song et al., 2020a) on the image label guidance task.

|  | RUNTIME (SECOND) | MEMORY (MB) |
| --- | --- | --- |
| SDG♣ w/o Stein correction | 602 | 18728 |
| SDG ($\alpha(t) > 0$, $\epsilon(t) > 0$) | $820_{(\uparrow+36.2\%)}$ | $18792_{(\uparrow+0.3\%)}$ |

