# OpenReview forum: "Stein Diffusion Guidance: Training-Free Posterior Correction for Sampling Beyond High-Density Regions"
_ICML.cc/2026/Conference — ICML 2026 regular_

### Official Review · Reviewer_sQqq · 2026-03-07

**Soundness:** 3
**Presentation:** 4
**Significance:** 2
**Originality:** 3
**Overall Recommendation:** 4
**Confidence:** 2

**Summary:**

The paper introduces Stein Diffusion Guidance (SDG), a diffusion guidance method that corrects the bias of Tweedie-based posterior approximations using a Stein variational posterior correction derived from a SOC formulation. The correction term improves reliability of SDG in low-density regions without introducing heavy computational cost of running full SOC. Experiments on image and molecular docking tasks show consistent gains over existing training-free methods.

**Compliance With Llm Reviewing Policy:**

Affirmed.

**Key Questions For Authors:**

1. The method roughly doubles runtime compared to standard training-free guidance. Do the authors envision strategies for reducing this overhead, or settings where the additional cost may outweigh the benefits?
2. SVGD uses RBF kernel as standard. The kernel should depend on the application geometry. Do you have a sense of kernel choice or hyperparameter sensitivity? It might be useful to include a discussion on these hyperparameters.
3. It is possible I missed it, but either way, could you please clarify for me whether you can provide a guarantee that the proposed Stein updates converge to the true posterior distribution in the diffusion setting?

**Limitations:**

See above of weaknesses and questions to the authors.

**Strengths And Weaknesses:**

Strengths:
- The paper is very well written with clear presentation of their diffusion posterior correct method
- The combination of Tweedie initialisation with Stein variational posterior correction is logically derived and well justified.
- Extensive experiments, especially in low-density molecular sampling, support the core claims and include useful ablations.
- Addresses instability and reward overestimation in diffusion guidance, a real issue in scientific generative modelling.

Weaknesses:
- Posterior score is approximated via MC approximation, may have high variability.
- The paper acknowledges that SVGD suffers instability in high-dimensional settings. The weaknesses of kernel-based Stein methods may limit scalability.

---

> ### Author Rebuttal · Authors · 2026-03-29
>
> We thank the reviewer for their valuable time and helpful feedback. We address the remaining concerns below.
>
> **W1: Posterior score approximation via Monte Carlo**
>
> Using Monte Carlo methods to approximate expectations is standard in statistical problems and is not expected to cause significant issues for SDG’s performance. In practice, one can use more samples or explore alternative sampling methods, such as importance sampling, to reduce estimation variance.
>
> **W2: Clarification on SDG stability and scalability**
>
> *Scalability*: SDG has $\mathcal{O}(N^2)$ computational complexity, similar to SVGD, due to kernel-based pairwise interactions between particles. However, the memory cost of these interactions is generally negligible compared to the memory required for model score evaluations. In practice, SDG can scale to a large number of particles with small memory overhead (e.g., 3000 particles for the molecule guidance task). We refer the reviewer to our response to Reviewer CJRr (W2+Q3) for a discussion on SDG’s scalability with respect to particle count.
>
> *Stability*: comparing Table 3 to Table 2, SDG consistently outperforms standard training-free diffusion guidance (SDG w/o Stein correction) across all particle count ablations. We refer the reviewer to our response to Reviewer TBXp (W1+Q1) for a discussion on SDG’s stability in high-dimensional settings.
>
> **Q1: Clarification on runtime overhead**
>
> For molecular drug discovery problems, which can span years of development across multiple stages, we believe that a near doubling of sampling runtime would not cause significant issues in practice. For applications requiring fast inference, such as image generation, we report a runtime overhead of around 36\% (Table 8, page 30) with DDIM solvers, which remains manageable and can be further reduced with faster solvers or by deploying SDG alongside few-step generation consistency models [1].
>
> **Q2: Robust SVGD's frameworks**
>
> Thanks for this interesting question. We agree that geometry-informed SGVD frameworks can enhance SDG's stability, and we highlight here some relevant perspectives: [2] introduces matrix-valued kernels together with Hessian- and Fisher information-based preconditioning matrices to adapt particle updates to the local geometry (curvature) of the target $p(x)$; LAWGD [3] formulates SVGD dynamics as a kernelized Wasserstein gradient flow of the Chi-squared divergence, in contrast to the (kernelized Wasserstein) gradient flow on the KL divergence in standard SVGD [4,5], leading to stronger convergence guarantees; and, E-SVGD [6] proposes equivariant kernels to better capture symmetries in structured data such as molecules. We will incorporate a discussion of robust geometry-informed SVGD variants for SDG in the revised manuscript.
>
> However, in all experiments, SDG performs well under the basic SVGD framework with an RBF kernel. We refer the reviewer to our response to Reviewer TBXp (Q3) for a discussion on SDG’s sensitivity to SVGD’s hyperparameters.
>
> **Q3: Clarification on convergence to the true posterior distribution in the diffusion setting**
>
> The Stein correction minimizes the KL divergence between the approximate posterior and the true diffusion posterior (term II of Equation 4), which is part of the derived surrogate stochastic optimal control objective (Proposition 3.2) for the reverse diffusion process $\mathbb{P}$ (Equation 1). Thus, in theory, the Stein correction updates guarantee that the approximate diffusion posterior samples converge to the true distribution.
>
> We thank the reviewer for their time and hope our responses address the concerns. We remain available for further questions if needed.
>
> *References*
>
> [1] (Song et al. ) Consistency Models. ICML 2023
>
> [2] (Wang et al.) Stein Variational Gradient Descent with Matrix-Valued Kernels. NeurIPS 2019
>
> [3] (Chewi et al.) SVGD as a kernelized Wasserstein gradient flow of the chi-squared divergence. NeurIPS 2020.
>
> [4] (Liu et al.) Stein Variational Gradient Descent: A General Purpose Bayesian Inference Algorithm. NISP 2016
>
> [5] (Qiang Liu) Stein Variational Gradient Descent as Gradient Flow. NISP 2017
>
> [6] (Jaini et al.) Learning Equivariant Energy Based Models with Equivariant Stein Variational Gradient Descent. NeurIPS 2021

---

> > ### Author Rebuttal · Reviewer_sQqq · 2026-04-02
> >
> > Thank you to the authors for their responses and clarifications. I would recommend including some of the clarifications in the paper to improve readability. My original score stands.

---

> > > ### Author Response · Authors · 2026-04-02
> > >
> > > We are pleased to hear that our responses have completely addressed the reviewer’s concerns. We would be grateful if the reviewer could kindly reconsider the scores in light of our responses and clarifications. We will include the necessary discussions to further improve the manuscript. We sincerely thank the reviewer again for their valuable time and helpful feedback.

---

### Official Review · Reviewer_TBXp · 2026-03-08

**Soundness:** 3
**Presentation:** 4
**Significance:** 3
**Originality:** 3
**Overall Recommendation:** 5
**Confidence:** 4

**Summary:**

This paper introduces a training-free diffusion guidance method for posterior correction in settings where the target lies in relatively low-density regions of the generative model. The paper argues that this setting is naturally connected to annealing: rather than viewing annealing as merely a heuristic to smooth out transitions during sampling, it is interpreted as a principled way to move through low-density regions where standard guidance can become unreliable. Building on this perspective, the method combines diffusion guidance with a Stein-based correction mechanism that improves posterior sampling without requiring additional model training. The paper provides a clear theoretical motivation for the method, derives the resulting algorithm clearly, and supports the approach with numerical experiments on guided generation tasks, including settings where low-density behaviour is especially important.

**Compliance With Llm Reviewing Policy:**

Affirmed.

**Final Justification:**

I appreciate the authors' responses to my questions and for their further clarifications. My assessment of the paper has not changed during the review period. I still believe that this is a strong paper and a nice contribution to the area. I think the paper is sufficiently original, sound and significant for ICML and therefore I would recommend Acceptance.

**Key Questions For Authors:**

1. Can you say a bit more about how the method scales as the ambient dimension increases, and where would the authors expect the Stein correction to become a bottleneck in practice?

2. Can the authors provide more intuition for when the Stein correction gives the largest benefit relative to standard annealed guidance?

3. How sensitive is the method to the particle count, kernel choice, or other Stein-specific hyperparameters?

**Limitations:**

Yes

**Strengths And Weaknesses:**

Strengths

Soundness: The paper is technically sound. The motivation is well articulated, the methodology is well aligned with the problem being studied, and the paper makes a convincing case for why standard guidance may struggle in low-density regions and why a Stein-based correction could be helpful. I quite liked the numerical results, they are very clear and supportive of the main claims. In particular, the paper does a good job of connecting the methodological design choices to the specific problem of sampling beyond high-density regions, rather than presenting the method as an isolated algorithmic trick.

Presentation:  This is one of the strongest aspects of the paper. The manuscript is very well written, the narrative is easy to follow, nice figures to illustrate the problem and methodology and the motivation is very clear. The interpretation of annealing as low-density sampling is especially nice. I also found the numerical section clearly explained and easy to interpret. Overall, the exposition is strong and helps make a fairly technical method quite accessible.

Significance: The paper addresses a meaningful problem. Training-free guidance methods are useful in practice, but their behaviour in low-density or more discovery-oriented regimes is a genuine challenge. The paper’s perspective could be valuable beyond the specific method that is presented here, because the reframing of annealing as low-density sampling gives a cleaner conceptual handle on an issue that arises in many generative inference settings. Even if the immediate impact is somewhat specialised, I can see the ideas being useful to researchers working on guided diffusion, posterior correction, and sampling methods for structured or scientific generation problems.

Originality:  I think the paper is sufficiently original. The novelty is not just in the algorithmic combination itself, but also in the way the paper conceptualises the problem. The framing of annealing as a low-density sampling mechanism is insightful, and the integration of a Stein correction into this setting is well motivated rather than arbitrary. This is a good example of originality arising from a thoughtful synthesis of ideas and from a better interpretation of why a practical mechanism works.

Weaknesses

Soundness:  My main technical reservation is about scalability. Because the method relies on a Stein-based particle correction, a natural concern is how well it scales with dimension and whether the correction remains effective in higher-dimensional regimes. This is a common issue for Stein methods more broadly, and it seems like the most likely practical limitation here as well. I do not view this as a flaw in the paper’s core argument, but it is an important caveat on how broadly the method is likely to be deployed.

Presentation:  The presentation is strong overall, so I do not have major criticisms here. If anything, the paper could benefit from making the scalability caveat more explicit in the main discussion so that readers immediately understand the likely trade-off, i.e., stronger posterior correction at the cost of a particle-based method whose efficiency may degrade in high dimensions.

Significance:  The contribution is meaningful, but the practical impact may depend on the regime of application. If the method is primarily most effective in moderate-dimensional or structured settings, then its influence may be strongest in specialised domains rather than as a universally applicable guidance strategy. For example, maybe guidance for small molecules. I still view this as a significant contribution, but I think the paper’s impact is likely to be shaped by how robustly the method scales.

Originality:  The paper is clearly original in perspective and method design, though some of the individual ingredients come from existing lines of work in diffusion guidance, annealing, and Stein methods. The originality therefore lies more in the synthesis, interpretation, and resulting algorithmic construction than in inventing an entirely new setup. I think that's fine, but it is worth stating precisely.

---

> ### Author Rebuttal · Authors · 2026-03-29
>
> We sincerely thank the reviewer for the detailed, constructive, and encouraging feedback. We address the remaining questions below.
>
> **W1+Q1: Clarification on SDG's scalability with respect to ambient data dimensions**
>
> Thanks for raising this question. We report the ambient data dimensions for each guidance task: molecule guidance (Node: 38x9, Edge: 38x38), label guidance (32x32), image deblurring (256x256), image super-resolution (256x256), and text-to-image style transfer (512x512). As observed, SDG scales effectively across a range of ambient data dimensions.
>
> In higher-dimensional settings, SDG may exhibit some instability due to the use of standard SVGD with heuristic kernel bandwidth RBF. More stable SVGD variants with adaptive kernels and bandwidth tuning can further improve SDG's performance in such cases.
>
> We agree that this raises an important point and will include the scalability discussion with respect to ambient dimensions in the revised version. We refer the reviewer to our response to Reviewer CJRr (W2+Q3) for a discussion on SDG's scalability with respect to particle count.
>
> **W2+Q2: Comparison to standard annealed guidance**
>
> The Stein correction ensures that the approximate posterior stays close to the true diffusion posterior, even in low-density regions, for high-reward guidance. In Section C.3.5 (Appendix, pages 27-28), Figures 20–22 compare SDG with standard annealed guidance (SDG w/o correction) under different low-density annealing levels. As observed, SDG maintains the meaningful structural (NSPDK $\downarrow$) and chemical (FCD $\downarrow$) distances to the data distribution, and generates significantly more valid molecules even under extreme low-density settings ($\alpha_{max}=0.5$). In contrast, standard annealed guidance largely drifts from the data distribution even at moderate annealing levels  ($\alpha_{max}=0.3$). This demonstrates that SDG provides the best benefit for low-density scientific discovery problems.
>
> **Q3: Sensitivity to SVGD's hyperparameters**
>
> In all experiments, we employ an RBF kernel with a median bandwidth, following the original SVGD framework [1]. Table 3 shows the ablation study results for different particle counts. Under this setting, varying the number of particles can lead to slight performance differences. However, all ablation results still outperform standard training-free guidance (SDG w/o Stein correction) in Table 2. We expect that more stable SVGD variants will further improve SDG’s stability in practice, i.e., ensuring that increasing the number of particles consistently improves performance. We refer the reviewer to our response to Reviewer sQqq (Q2) for a discussion on geometry-informed SVGD variants, which have the potential to further reduce SDG’s sensitivity to hyperparameter choices.
>
> We are pleased that our contributions are well-received, and we hope our responses have addressed the remaining questions. We would be happy to provide further clarification if helpful.
>
> *References*
>
> [1] (Liu et al.) Stein Variational Gradient Descent: A General Purpose Bayesian Inference Algorithm. NISP 2016

---

> > ### Author Rebuttal · Reviewer_TBXp · 2026-04-01
> >
> > Thank you to the authors for responding to my comments. The answers provided adequately address my questions and I hope the authors will incorporate these answers in a revised version of their paper.
> >
> > On reflection, I feel that my score is appropriate and I will maintain it.

---

> > > ### Author Response · Authors · 2026-04-01
> > >
> > > We are glad that our responses have addressed the remaining concerns. We thank the reviewer for their valuable time and helpful feedback. We will ensure that the relevant discussion is included in the final version.

---

### Official Review · Reviewer_CJRr · 2026-03-11

**Soundness:** 3
**Presentation:** 3
**Significance:** 3
**Originality:** 4
**Overall Recommendation:** 5
**Confidence:** 4

**Summary:**

This paper introduces the notion of Stein Diffusion Guidance (SDG), which is a training-free steering approach for diffusion models, which addresses the limitations of Tweedie's formula based approaches, particular focused on settings where highly valuable samples may lie in low-density regions.

To this end, the authors start from a novel Stochastic Optimal Control problem targeting low density regions (equiv penalising high density regions), then relax the associated value function variationally.   This yields a decomposition into three terms, the last of which is a KL regulariser between the controlled and original conditional distributions, keeping the system from diverging too far away.

Based on this the authors propose a surrogate SOC problem, whose associated optimal control u^* leverages a kernelised Stein gradient to main the regularisation.    This is a significant approximation, but yields reasonable results numerically.   Even this control is expensive, so they further approximate the scheme with a back-and-forth approach (map to X_T via tweedie, correct, then map back via sampling).

In some sense, SDG offers a middle ground between Tweedie approaches, where a posterior sample x_T is approximated from a noisy sample x_t,  and far more heavy stochastic optimal control (Richter, Nusken, etc) approaches, which require deep backpropagation, etc.

All in all, this approach is demonstrated experimentally on various tasks including image guidance and molecular docking and shown to be quite effective.

**Compliance With Llm Reviewing Policy:**

Affirmed.

**Final Justification:**

I believe the authors have understood the issues i have highlighted in the soundness, and will address them.  Therefore I re-evaluate my soundness score, and overall score accoridngly

**Key Questions For Authors:**

1. Could you clarify the induced marginal density in Proposition 3.1 and spell out what it actually is?  In its current form it appears to depend on the terminal density.  This is quite important as the rest of the construction depends on this object.

2. The stein correction depends on $\nabla_{x_T}\log p(X_T | x_t) \approx s_{\theta}(x_T) - \eta(t)s_{\theta}s_{\theta}(x_t)$.  The paper doesn't really address the accuracy of this approximation, particularly in regimes which are relevant, namely low density settings.  Can the authors shed some light on this?

3. Can the authors discuss the computational complexity of the Stein Step in terms of number of particles?  A more explicit discussion on the $N$ dependence would help understand which regimes its reasonable to use SDG.

4. Can the authors be more clear about the relationship between the gradient of KL and Stein gradient descent.   This is clearly an approximation, and it should be acknowledged more clearly, and ideally addressing the limitations of this (which the numerical experiments already do to some extent).

I'd be delighted to increase my score if these could be carefully addressed.

**Limitations:**

Yes

**Strengths And Weaknesses:**

Soundness:   Generally good, but not completely solid.   The reinterpretation of the training-free guidance through surrogate stochastic optimal control is nice.   Similarly, the experiments offer strong evidence that this approach is effective.   However many liberties are taken in the derivation of the process, which maybe need to be drawn out explicitly in the text.

One of the biggests conceptual gaps is the fact that the direction of steepest descent of KL is not automatically Stein gradient descent.  This is only true if you are searching over a limited set of directions, namely pushforwards of the form (I + epsilon Phi)_# mu, where Phi is an element of the unit ball of H^_d for some RKHS H = H(k).    This is a very significant approximation, but in the current form of the paper appears as if it is an exact equality, or even a close approximation.     In particular, this approximation breaks down spectactularly as the dimension increases, something which the authors seem to acknowledge in the ablation studies.   Similarly other claims are made that need some clarification.  In proposition 3.1, $p_t^u$ is the induced marginal density, but this clearly depends on $x_T$ which is a random variable, and thus its unclear what is meant here.

Also slightly hidden under the hood is the complexity of the O(N^2) calculations required for every step of the Stein correction, due to the kernel (or at least I could not find reference to this)  This is quite significant, especially given we're looking at numerical experiments in the order of N=3000.   This should have been drawn out more.

Presentation:   The presentation is generally good, and the structure of the paper is well-organised.  The main weakness in the presentation is in the theory which was a bit loose at times.   The numerical experiment section was very solid and well written.

Significance:  This is clearly a significant paper with high potential for impact, based on the numerical experiments.   Particularly, the molecular experiments where sample discovery in low-density regions is clearly important.   The paper also forms a very nice story in bridging Tweedie-based approaches and SOC based approaches for training-free guidance.

Originality: To my knowledge, this appears to be an original idea, which introduces a genuinely new method and conceptual approach.

---

> ### Author Rebuttal · Authors · 2026-03-29
>
> We thank the reviewer for their thoughtful evaluation of our paper and for the constructive technical comments. We carefully address the concerns below.
>
> **W1+Q4: Clarification on the steepest KL descent  direction and Stein gradient descent**
>
> Thanks for raising this important point. In Lemma 2.1, we explicitly state that the steepest KL descent direction is defined as the maximizer of the Stein discrepancy over the unit ball of the RKHS  $\mathcal{H}(k)$, which corresponds to Stein gradient descent $ \phi^* $  s.t. $|| \phi^* ||_{\mathcal{H}} \leq 1$. Its closed-form solution is given by kernelized Stein discrepancy (lines 112–116).
>
> Moreover, our surrogate control objective recovers exactly this direction: specifically, Equation 4 (term II, line 234) corresponds to finding the steepest KL descent direction, which yields the Stein gradient descent (Equation 5) by applying Lemma 2.1 and $\phi^*$ (lines 112-116). However, directly computing this optimal descent direction is computationally expensive in high-dimensional settings (lines 258–262, 1140–1144).
>
> To address this, we propose a back-and-forth Stein correction that instead computes the steepest KL descent direction on $\mathcal{M}_T$ (lines 271-274), leading to the new optimal direction (Equation 6) by again applying Lemma 2.1 and $\phi^*$ (lines 112-116).
>
> **W2+Q1: Clarification on the induced marginal density in Proposition 3.1**
>
> In Proposition 3.1, we apply the classical SOC result from Lemma 2.2 with our proposed cost functional for low-density sampling. The induced density $p^u_t(x_t)$ appears in both the value function $V$ and the optimal controller $u^*$, which involve the expectations under the uncontrolled process $\mathbb{P}$ (Equation 1).
>
> To answer the question, we have that the controlled density $p^u_t(x_t)$ depends on the terminal sample $x_T$ (with its properties evaluated by the reward function $r$), which is obtained by simulating the uncontrolled process $\mathbb{P}$ from $x_t$ (as stated in lines 214-217, Lemma 2.2, and more generally in Section 2.3). However, this simulation is computationally expensive in a standard SOC. In this work, we avoid this simulation step by using a one-step posterior approximation via Tweedie’s formula, combined with our proposed Stein posterior correction.
>
> **W2+Q3: The computational complexity of SDG in terms of the number of particles $N$**
>
> We agree that SDG inherits the computational bottleneck of SVGD, with complexity $\mathcal{O}(N^2)$, due to pairwise interactions among particles. Even though we employ the RBF kernel, whose gradients admit a closed form, the overall complexity of SDG remains $\mathcal{O}(N^2)$.
>
> However, in our experiments, the memory cost of RBF pairwise particle interactions is generally negligible compared to the memory required for model score evaluations. Liu et al. [1] also observed the dominant computational cost from data gradient evaluations, which are similar to our model score evaluations (please refer to “Complexity and Efficient Implementation” [1]).
>
> Table 6 (page 28) and Table 8 (page 30) report the memory consumption for the molecule guidance task (with $N=3000$) and for the image label guidance task (with $N=256$), respectively. In both cases, we observe only a small memory overhead of SDG compared to its counterpart w/o the Stein correction. Therefore, despite the $\mathcal{O}(N^2)$ complexity of kernel pairwise computations, applying SDG to general diffusion guidance tasks does not appear to impose a significant memory burden in practice.
>
> We agree that this question raises a valid point and will include the discussion in the revised version.
>
> **Q2: Clarification on Lemma 3.3**
>
> The result of Lemma 3.3 is necessary for evaluating the optimal direction $\phi^*(x_T)$ in Equation 6. The approximation arises from using a one-sample Monte Carlo estimate for the expectation in line 1071 and in Equation 14 (proof of Lemma 3.3). Notably, the conditional posterior score approximation $\nabla_{x_T} \log p(x_T|x_t)$ depends only on the model score $s_{\theta}(\cdot)$ and does not depend on the low-density annealing schedule $\alpha(t)$. Therefore, this approximation is agnostic to the data regime (high- or low-density regions). Consequently, the accuracy of this approximation is determined primarily by the quality of the pretrained model $s_{\theta}(\cdot)$.
>
> We thank the reviewer for their time and hope our responses address the concerns. We remain available for further questions if needed.
>
> *References*
>
> [1] (Liu et al.) Stein Variational Gradient Descent: A General Purpose Bayesian Inference Algorithm. NISP 2016

---

> > ### Author Rebuttal · Reviewer_CJRr · 2026-04-02
> >
> > Regarding Steepest Descent of KL: This may be a language issue however, the claim in Lemma 2.1 is "The steepest descent direction of the Kullback Leibler (KL) divergence ... is the function ... that maximizes the Stein Discrepancy over the unit ball
> > of the Reproducing Kernel Hilbert Space".       This is not necessarily true - this is a choice of tangent space that you have explicitly chosen.    For example you could have chosen steepest descent over Wasserstein, Fisher-Rao, or MMD, etc.   The way you have written it now, it sounds like it is always canonically SD.
> >
> > In your response you state that "you have defined the steepest descent in this way", however, this does not come across in the text.
> >
> > Regarding the induced marginal density:  My point is that the marginal density for x_t cannot depend on x_T.   The correct marginal density would be the conditional expectation of this quantity.    The mathematics checks out all the same but the terminology you are using for this term is incorrect.
> >
> > - EDIT - Based on the response provided to this comment, I feel I am finally aligned with the authors on this.  For that reason I will update score accordingly.   If accepted, I really hope that the authors do carefully take this feedback on board, because this is crucial for clarity and correctness.

---

> > > ### Author Response · Authors · 2026-04-03
> > >
> > > We thank the reviewer for the additional clarifications:
> > >
> > > 1. We agree that *steepest descent* is geometry-dependent. In Lemma 2.1, the claim is intended to characterize the direction of steepest descent of KL *restricted to RKHS transport perturbations*, rather than a canonical statement about KL. We will revise the wording and clearly acknowledge that SDG optimizes the KL term only within this restricted class of perturbations.
> > >
> > > 2. SDG mainly relies on the SOC quantities $V$ and $u^*$, both of which involve $p_t^u(x_t)$ through expectations under $\mathbb{P}$ (Proposition 3.1). We agree with the reviewer that, if interpreted as a marginal density, this term should not depend explicitly on $x_T$. The correct form is obtained by integrating out the terminal variable under $\mathbb{P}$, i.e, $p_t^u(x_t)=p_t^{1-\alpha(t)}(x_t)\mathbb{E_{\mathbb{P}}}[\exp(\beta(t) r(x_T))|x_t]$. We will revise the manuscript accordingly to ensure correct terminology.
> > >
> > > We thank the reviewer for their thoughtful comments, which improve the clarity and correctness of our formulation.
> > >
> > > **[Updated]** We are pleased to hear that our responses have adequately addressed the reviewer’s concerns. We appreciate the reviewer raising the score to reflect these improvements. We will incorporate the relevant discussion from the rebuttal to ensure the method’s clarity and correctness in the revised version.

---

### Official Review · Reviewer_ZHeS · 2026-03-12

**Soundness:** 2
**Presentation:** 4
**Significance:** 3
**Originality:** 3
**Overall Recommendation:** 5
**Confidence:** 4

**Summary:**

The authors provide a novel training-free approach for guiding diffusion models that takes inspiration from stochastic optimal control (SOC). In order to optimize with SOC, backpropagation needs to be done through the whole sequential process, which can be very costly. To avoid this, one can use Tweedie's formula that allows one-shot denoising. However, the authors show that this leads to biased results and does not work well in low-density regions. As such, they introduce Stein Diffusion Guidance, with a correction term that is based on samples to fix the tradeoffs between SOC and Tweedie.

**Compliance With Llm Reviewing Policy:**

Affirmed.

**Final Justification:**

Initially, I had some doubts about this paper, especially because of the limited related work section and some limited experiments.

While carefully considering the paper again, I think that the quality of the overall paper, especially after considering the other reviews, is high and in my opinion the paper should be accepted.

Especially the additional analysis of the related work and them clarifying the distinctions has convinced me that their ideas are novel and beneficial for the community.

As such, the authors have resolved my initial doubts and I will recommend "accept".

**Key Questions For Authors:**

1. When the authors introduce the variational bound in Proposition 3.2, it sounds like the expectation and the KL divergence together recover the classical training-free diffusion guidance. In 3.3 however, it sounds like only the expectation part is the standard training-free diffusion guidance with the KL being introduced in this work. Could the authors clarify this point?
2. The paper generally investigates the case where we do not have / need a time-dependent reward function but have only one classifier for the final result. In some cases, we can already estimate a guidance score on noisy states / intermediate samples. When access to a time-dependent reward function is given, do you think that your method could still offer some benefits?
3. It is not particularly clear to me how the samples for the Stein component are obtained. I suspect that high-quality samples of low-probability regions can improve the quality, but how are they generated? Are they picked from the training data? Are they just random samples generated with a diffusion model? Does it make sense to also include noisy samples?
4. On the note of samples, the authors say that the Stein should in principle work better with more samples, where they note some inconsistencies. Could the authors elaborate why this is happening? Did they observe a double descent problem, where more sample initially make the algorithm perform worse and then better again?
5. Did the authors do any ablations on $\alpha$? in the appendix I found that there is a linear and a constant version, but it is not particularly clear to me how one would choose it and how much impact it has on the algorithm.
6. Could the authors elaborate on how the three proteins were selected? I am surprised to only see these three systems and wondering if they are cherry-picked. Results aggregated over a larger number of proteins would hugely increase my confidence in this approach.

**Limitations:**

I think for the most part, the limitations are well-addressed, but I would like to see the authors putting their work into context with more existing solutions (e.g., related work section), highlighting differences and limitations.

**Strengths And Weaknesses:**

## Strengths
- The authors present a well-written manuscript with a very nice introduction to the field, and great preliminaries that explain the core principles behind SOC and Stein variational inference.
- As to my knowledge, the authors propose a novel approach to fix the drawbacks of SOC and Tweedie's formula.
- The authors evaluate their approach for different systems: Image generation and small-ligand docking. Two well-studied problems with potentially a huge impact across many disciplines.
- In some experiments, we can see that their Stein correction can lead to a huge impact, e.g., in the novel hit ratio.

## Weaknesses
While I think that the presentation of this paper is excellent, in my opinion there are some weaknesses with respect to soundness, evaluation, and depth of experiments.

- One major point that makes it a bit challenging to put the work into context is the lack of a related work section. While the authors do a good job in presenting the theoretical background (SOC, Stein etc.), the work of "diffusion guidance" is so vast that I was expecting a related work section that at least puts other works into context. For instance [1] is a well-known work in this field and there are also recent high-profile publications [2] that make use of it. (I am aware that [2] was published after the ICML deadline). I do not expect the authors to benchmark against 20 different Guidance protocols, but listing some crucial ones would be great [1, 3, 4...]. Also, I was surprised not so see citations such as [5], which are widely known.
- While the experiments generally show that the proposed method improves over the state of the art, the results are often mixed for images and not always consistent. More on experiments in questions.
- I think using the different symbols for the different variations of SDG is very confusing, and I had to check multiple times what each symbol corresponds to. I think the authors should try to introduce $\alpha$ and $\epsilon$ a bitt sooner and maybe with some more intuitive notions so that the four different variants are easier to digest quickly.
- I am not sure if Figure 2 adequately describes the process. The authors describe that their approach maps forward and backward, I do not see any arrows going in the other direction. Maybe I misunderstood something in this figure, but I think at least the caption could be improved.

[1] Skreta et al., 2025, Feynman-Kac Correctors in Diffusion: Annealing, Guidance, and Product of Experts

[2] Yu et al., 2026, Enhanced Diffusion Sampling: Efficient Rare Event Sampling and Free Energy Calculation with Diffusion Models

[3] Sadat et al., 2024, No Training, No Problem: Rethinking Classifier-Free Guidance for Diffusion Models

[4] Tang et al., 2025, Diffusion Models without Classifier-free Guidance

[5] Ho et al., 2022, Classifier-Free Diffusion Guidance

---

> ### Author Rebuttal · Authors · 2026-03-30
>
> We thank the reviewer for their helpful comments. Due to the character limit, we opt to address the key concerns below.
>
> **W1: Missing related works**
>
> - *Training versus inference*: [4,5] focus on CFG-based training from scratch, which fundamentally differs from our *inference-time-only* setting for pretrained diffusion models.
>
> - *Guidance on noisy samples $x_t$ versus posterior samples $x_T$*: [1,2] employ weighted-sampling reward guidance on noisy states $x_t$ via SMC, requiring conditional pretrained models or classifiers trained on noisy samples. [3] aims for similar sample quality as CFG using simpler approaches (e.g., ICG by perturbing condition labels and TSG by perturbing time conditions). In contrast, SDG solves the guidance problem on diffusion posterior samples $x_T$ with off-the-shelf differentiable reward functions, dubbed *training-free guidance (TFG)*.
>
> We will clarify these points in the revised version.
>
> **W2: The symbols $\alpha$ and $\epsilon$**
>
>  We define these symbols in context: $\alpha(t)$ is the low-density schedule (Section 3.1, line 192), and $\epsilon(t)$ is the particle update stepsize (Section 3.3, line 283; see also Algorithm 1). We also summarize all symbols in Table 2 for clarity.
>
> **Q1: Proposition 3.2**
>
> In Section 3.2 (lines 203–219), we describe the role of each term in Proposition 3.2 within the surrogate SOC variational bound. In particular, only *the expectation* (the term in line 194 and its decomposition in line 198) corresponds to standard training-free guidance, where posterior samples are approximated via Tweedie’s formula.
>
> **Q2: Comparison with classifier-based guidance and $x_t$-based methods**
>
> The advantages of posterior guidance on $x_T$  stem from practical considerations: it enables direct use of *off-the-shelf* classifiers or reward functions without additional training. Moreover, in many scientific settings, e.g., inverse problems [6], the inputs only consist of posterior samples, making SDG broadly applicable beyond classifier-based guidance. As noted in Section 4.2 (lines 312–315), SDG still outperforms the classifier-based method MOOD on 2/3 target proteins. We also discuss the limitations of classifier-based guidance (lines 47–59).
>
> **Q3: The inputs to the Stein component**
>
> We refer the reviewer to Algorithm 1 (Appendix, page 18) for a step-by-step description of SDG. Sampling is initialized from a standard Gaussian diffusion prior (line 940), i.e., not from training data. The Stein component operates on diffusion posterior samples approximated via Tweedie’s formula (line 943), and does not include noisy samples $x_t$.
>
> **Q4: The performance w.r.t. different particle counts**
>
> We observe some variability in performance across particle counts (e.g., for the Jak2 receptor in Table 3). This likely stems from limitations of standard SVGD, as also noted in our ablation study on particle counts (line 433).
>
> Importantly, across all particle settings in Table 3, SDG consistently outperforms standard TFG (SDG w/o Stein correction) in Table 2. For image-guidance tasks, we fix the number of particles per problem (w/o ablation) and still observe strong performance. Further gains may be achieved by tuning the particle count or adopting improved SVGD variants.
>
> We refer the reviewer to our responses to Reviewer ZHeS (W2+Q3), TBXp (Q3), and sQqq (Q2) for discussions on scalability, stability, and robust SVGD variants, respectively.
>
> **Q5: Ablations on $\alpha$**
>
> We select the low-density annealing schedule that yields the best performance (Table 4, page 25). Appendix C.3.5 (Figures 20–22) reports ablations of SDG in terms of structural (NSPDK $\downarrow$), chemical (FCD $\downarrow$) distances, and molecular validity, across different low-density levels. Overall, SDG preserves meaningful distances to the data distribution and produces substantially more valid molecules at extreme low-density levels ($\alpha_{\max}=0.5$) compared to standard TFG (SDG w/o Stein correction), further confirming SDG's effectiveness in low-density sampling.
>
> **Q6: The molecule experiment.**
>
> Due to the high computational cost of docking simulations in our academic lab, we focused on a limited set of proteins. Here, we include an additional target, **PARP1** (Poly[ADP-ribose] polymerase-1); the results on the novel hit ratio(\%) are reported below.
>
> HierVAE: 0.553 | MORLD:  0.047 | FREED: 3.627 | GDSS: 1.933 | MOOD: 7.017 |  **SDG: 8.780** | SDG w/o Stein correction (standard TFG): 0.671
>
> *Robust evaluation protocol*: 3000 molecules per target protein, mean +- std over 3 runs across multiple metrics: NSPDK, FCD, Novelty, Uniqueness, Validity, SA, QED, DS, and novel hit ratio.
>
> Importantly, SDG significantly outperforms standard TFG (SDG w/o Stein correction) across all the proteins.
>
> We thank the reviewer for their time and remain available for further questions if needed.
>
> *References*
>
> [6] (Chung et al.) Diffusion Posterior Sampling for General Noisy Inverse Problems. ICLR 2023

---

> > ### Author Rebuttal · Reviewer_ZHeS · 2026-04-02
> >
> > I would like to thank the authors for their responses and further clarifications.
> >
> > I especially appreciate their commitment to revising and expanding on the related work and adding an additional protein target.
> >
> > As such, I would like to positively adjust my score.

---

> > > ### Author Response · Authors · 2026-04-02
> > >
> > > We sincerely thank the reviewer for their valuable time and constructive feedback. We are grateful that the reviewer has updated the score to reflect the improvements. We will ensure that the suggested works are appropriately discussed in the 'Related works' section of the final version.

---

### Decision · Program_Chairs · 2026-04-30

**Decision:**

Accept (regular)

**Comment:**

This paper proposes Stein Diffusion Guidance (SDG), a training-free diffusion guidance method that uses a Stein variational correction derived from a stochastic optimal control formulation. The method addresses limitations of Tweedie-based posterior approximations in low-density regimes by introducing a particle-based correction that improves alignment with the true diffusion posterior. Experiments on image guidance and molecular docking tasks show consistent gains over prior training-free methods.

All four reviewers are positive, noting that the paper is well-written, clearly motivated, and empirically strong. The method is viewed as a meaningful combination of SOC and Stein variational inference with strong practical performance, especially in scientific and low-density sampling tasks.

Concerns mainly relate to theoretical clarity and scalability, including the assumptions behind the link between KL descent and Stein updates, particle complexity, Monte Carlo approximation noise, and limited discussion of convergence guarantees and kernel sensitivity. The rebuttal clarifies several points and provides additional discussion, but some theoretical and scalability issues remain.

Overall, the paper is considered a solid and practically impactful contribution to training-free diffusion guidance, with strong empirical results despite remaining theoretical and scalability limitations.